# Health and social behaviour through pandemic phases in Switzerland: Regional time-trends of the COVID-19 Social Monitor panel study

**André Moser**[1,2], **Viktor von Wyl**[2], **Marc Höglinger**[3]*

**1** CTU Bern, University of Bern, Bern, Switzerland, **2** Epidemiology, Biostatistics and Prevention Institute, University of Zurich, Zurich, Switzerland, **3** Winterthur Institute of Health Economics, Zurich University of Applied Sciences, Winterthur, Switzerland

* marc.hoeglinger@zhaw.ch

**Data Availability Statement:** All relevant data are within the manuscript and its Supporting Information files.

**Funding:** The Social Monitor study has received funding from the Swiss Federal Office of Public

## Abstract

### Background

Switzerland has a liberal implementation of Coronavirus mitigation measures compared to other European countries. Since March 2020, measures have been evolving and include a mixture of central and federalistic mitigation strategies across three culturally diverse languages regions. The present study investigates a hypothesised heterogeneity in health, social behavior and adherence to mitigation measures across the language regions by studying pre-specified interaction effects. Our findings aim to support the communication of regionally targeted mitigation strategies and to provide evidence to address longterm population-health consequences of the pandemic by accounting for different pandemic contexts and cultural aspects.

### Methods

We use data from from the COVID-19 Social Monitor, a longitudinal population-based online survey. We define five mitigation periods between March 2020 and May 2021. We use unadjusted and adjusted logistic regression models to investigate a hypothesized interaction effect between mitigation periods and language regions on selected study outcomes covering the domains of general health and quality of life, mental health, loneliness/isolation, physical activity, health care use and adherence to mitigation measures.

### Results

We analyze 2,163 (64%) participants from the German/Romansh-speaking part of Switzerland, 713 (21%) from the French-speaking part and 505 (15%) from the Italian-speaking part. We found evidence for an interaction effect between mitigation periods and language regions for adherence to mitigation measures, but not for other study outcomes (social behavior, health). The presence of poor quality of life, lack of energy, no physical activity,

Health and from Health Promotion Switzerland. The funders had no role in study design, data collection and analysis, decision to publish, or preparation of the manuscript.

**Competing interests:** The authors have declared that no competing interests exist.

health care use, and the adherence to mitigation measures changed similarly over mitigation periods in all language regions.

## Discussion

As the pandemic unfolded in Switzerland, also health and social behavior changed between March 2020 to May 2021. Changes in adherence to mitigation measures differ between language regions and reflect the COVID-19 incidence patterns in the investigated mitigation periods, with higher adherence in regions with previously higher incidence. Targeted communcation of mitigation measures and policy making should include cultural, geographical and socioeconomic aspects to address yet unknown long-term population health consequences caused by the pandemic.

## Introduction

Europe faced the second wave of the Coronavirus pandemic during the autumn and winter months of 2020. Switzerland is among the countries with the highest case numbers and deaths per capita (https://coronavirus.jhu.edu, accessed December 29, 2020). Non-pharmaceutical mitigation measures such as social distancing, testing, or restricting mobility can substantially reduce Coronavirus transmission [1, 2]. Switzerland has a liberal implementation of mitigation measures to slow down Coronavirus transmission compared to other European countries. These mitigation measures center around self-responsibility. Freedom of movement is not restricted, and shops, businesses, and schools have remained open, while restaurants were forced to close only shortly before the Christmas holidays [3]. The Oxford COVID-19 Government Response Tracker Stringency Index for November 2020 was 37.5 for Switzerland, 60.7 for Germany, 66.7 for Italy and 78.7 for France, with a higher index indicating a more stringent implementation of mitigation measures (https://covidtracker.bsg.ox.ac.uk/stringency-scatter, accessed December 29, 2020).

Switzerland's mitigation strategy can be divided into different mitigation periods with stepwise increasing or decreasing stringency of measures and implementation at different political levels. The Swiss Federal Council coordinated nationwide and centralized mitigation measures (for example, the nationwide lockdown on March 16, 2020) until June 19, 2020, when the state of emergency as per the Swiss Epidemic Law ended. Thereafter, the 26 cantonal authorities were mainly responsible for a federalistic implementation of mitigation measures and remained in charge until January 17, 2021 (but with close federal and intercantonal coordination). While the epidemic situation worsened in the autumn months, cantons reacted differently and with varying strength of mitigation measures. This led to a patchwork of heterogeneous mitigation measures within small spatial proximity, with restaurants in one canton being closed and others remaining open.

Switzerland's federalistic system overlaps with culturally diverse language regions. Citizens from the same language region often share common cultural traits, and health, risk, social and prevention behaviour differs between language regions in many regards [4–8]. For example, cultural differences in vaccination uptake were reported before the Coronavirus pandemic in Switzerland [8, 9]. The administered COVID-19 vaccination rate is highest in the Italian-speaking part of Switzerland and varies substantially between language regions (https://www.covid19.admin.ch/de/overview/, accessed, May 26, 2021). While regional and temporal variation in COVID-19 incidence patterns influence decision-making on the cantonal or

nationwide implementation of mitigation measures, the socio-cultural context may play an important role in communication, the awareness of the pandemic situation and the individuals' adherence to mitigation measures [10]. Because Switzerland is surrounded by European countries with varying mitigation strategies, the emerging pandemic challenged the different regions in Switzerland in several ways. Mitigation measures as well as other consequences of the pandemic (e.g. infection rates, widespread fears, emergency department crowding) likely impact a range of relevant public health and behavioral outcomes leading to different coping strategies over the phases of the pandemic and across language regions.

## Objectives and research hypotheses

Positive and negative consequences of implemented mitigation measures and the pandemic on changes in relevant health and social behavior, health care use and the population's adherence to mitigation measures during the Coronavirus pandemic have not been investigated in Switzerland so far. COVID-19 incidence patterns differ substantially between cantons and language regions, and heterogeneously-implemented mitigation measures may lead to different behaviors across language regions. We hypothesize that an interaction effect between pre-specified mitigation periods and language regions on behavioral changes exists. To investigate this research hypothesis, we use data from the COVID-19 Social Monitor, a population-based online survey which has longitudinally collected various aspects of social and health behavior since the beginning of the pandemic [11]. We analyze changes in these outcomes over the course of the pandemic. Our results provide first evidence about the extent of observable variations over time and about differences between the Swiss language regions. Our findings aim to support the communication of regionally targeted mitigation strategies and to provide evidence to address longterm population-health consequences of the pandemic by accounting for different pandemic contexts and cultural aspects.

## Methods

### Study population

Our study population covers a stratified random sample of a large cohort of the resident population in Switzerland with online access aged 18 years or older. Stratification was based on age, sex, and language region, i.e. the cohort is representative for Switzerland with respect to these three stratification criteria.

### Data source

We use the COVID-19 Social Monitor survey waves 1 to 16 (March 2020 to May 2021). In brief, the COVID-19 Social Monitor is a population-based online survey which collects relevant aspects for a broad range of domains over multiple survey waves [11]. Study participants have been sampled from an online-panel whose members have been actively recruited using random probability sampling based on national landline telephone directories and random digit dialling of mobile phone numbers. An initial survey sample (survey wave 1) of 2,026 participants was interviewed from March 2020 onwards in a total of 11 survey waves. In December 2020, an additional sample of 1,355 individuals participated in the survey. These were–together with the initial sample—interviewed in four subsequent survey waves. Survey participants were randomly drawn from age, gender and language region strata. Table 1 shows a schematic overview of the survey waves from March 2020 to May 2021. We use data from the Federal Office of Public Health (https://www.covid19.admin.ch/de/overview/, accessed, May 18, 2021) for cantonal new COVID-19 cases from February 24, 2020 (the first COVID-19 case

**Table 1. Schematic overview of COVID-19 Social Monitor survey waves.**

| Survey wave | 1 | 2 | 3 | 4 | 5 | 6 | 7 | 8 |
|---|---|---|---|---|---|---|---|---|
| Month | March 2020 | April 2020 | April 2020 | April 2020 | May 2020 | May 2020 | June 2020 | July 2020 |
| N of initial sample | 2026 | 2026 | 2026 | 2026 | 2026 | 2026 | 2026 | 2026 |
| N of additional sample | - | - | - | - | - | - | - | - |
| N of analysis sample | 2026 | 2026 | 2026 | 2026 | 2026 | 2026 | 2026 | 2026 |
| No. of participants | 2026 | 1537 | 1540 | 1729 | 1673 | 1616 | 1522 | 1508 |
| Non-participation (%) | 0 | 24% | 24% | 15% | 17% | 20% | 25% | 26% |
| **Survey wave** | **9** | **10** | **11** | **12** | **13** | **14** | **15** | **16** |
| Month | August 2020 | October 2020 | November 2020 | December 2020 | January 2021 | February 2021 | April 2021 | May 2021 |
| N of initial sample | 2026 | 2026 | 2026 | 2026 | 2026 | 2026 | 2026 | 2026 |
| N of additional sample | - | - | - | 1355 | 1355 | 1355 | 1355 | 1355 |
| N of analysis sample | 2026 | 2026 | 2026 | 3381 | 3381 | 3381 | 3381 | 3381 |
| No. of participants | 1532 | 1511 | 1492 | 2802 | 2564 | 2346 | 2219 | 2154 |
| Non-participation (%) | 24% | 25% | 26% | 17% | 24% | 31% | 34% | 36% |

in Switzerland) to May 03, 2021 (the latest interview date of survey wave 16). Because new cases are only reported on a cantonal level and language regions do not follow cantonal borders, we assign the canton Ticino to the Italian-speaking part, the cantons Fribourg, Geneva, Jura, Neuchâtel, Vaud and Valais to the French-speaking part, and the remaining cantons to the German/Romansh-speaking part of Switzerland. In order to plot maps, we used free geodata from the Federal Office of Topography swisstopo.

## Mitigation periods

Table 2 shows an overview of four a priori defined time periods from March 2020 to December 2020, based on the stringency of mitigation measures in Switzerland. The first period started from the date of the nationwide lockdown (March 16, 2020) and ended one day before the date of the nationwide reopening of stores and public schools (May 10, 2020). The second period lasted from May 11, 2020 to one day before the date of the mandatory nationwide implementation of face mask wearing in public transport (July 5, 2020). The third period started on July 6, 2020 and ended one day before the slowdown (i.e. mandatory nationwide wearing of face masks in public buildings, ban on spontaneous gatherings with more than 15 persons and recommended work from home) on October 18, 2020. The fourth period covers from October 19, 2020 to January 17, 2021 and was marked by the entry into a second pandemic wave with high case numbers. On January 18, 2021, the Swiss Federal Council announced a period with nationwide stringent mitigation measures with the closing of restaurants, mandatory homeoffice regulations and a ban on gatherings with more than 5 people in households (fifth period).

## Study outcomes

We use the following study outcomes grouped in six **domains of interest**. Study outcomes were a priori selected to cover a broad domain of relevant health and behaviorial aspects and were (mostly) consistently included in all survey wave questionnaires. All study outcomes stem from single questions which allowed for categorical answers (e.g. on a Likert-scale). The source and original question used in the survey questionnaire is provided in S1 Table. Study outcomes were dichotomized to communicate results in terms of proportions and odds ratios.

**Table 2. Overview of mitigation periods and implemented mitigation measures.**

| Period | Coordination level | Mitigation measures* |
|---|---|---|
| (1) March 16, 2020 to May 10, 2020 | Nationwide | Ban on gatherings >5 persons |
| | Nationwide | Public school closures |
| | Nationwide | Closure of stores and markets |
| | Nationwide | Closure of restaurants and bars |
| | Nationwide | Partial border closure |
| | Nationwide | Testing of symptomatic cases |
| | Nationwide | Hygiene rules, isolation, quarantine |
| (2) May 11, 2020 to July 5, 2020 | Nationwide | Ban on gatherings >30 persons |
| | Nationwide | Partial school closure |
| | Nationwide | Partial border closures |
| | Nationwide | Testing of symptomatic cases |
| | Nationwide | Hygiene rules, isolation, quarantine |
| (3) July 6, 2020 to October 18, 2020 | Nationwide | Face masks in public transport |
| | Nationwide | Ban on gatherings in public places >15 persons |
| | Nationwide | Allowance of mass gatherings >1000 persons |
| | Nationwide | Testing of symptomatic cases |
| | Nationwide | Hygiene rules, isolation, quarantine |
| | Cantonal | Face mask wearing in stores and public buildings |
| (4) October 19, 2020 to January 17, 2021 | Nationwide | Face masks in public transport |
| | Nationwide | Face mask wearing in busy places and public buildings |
| | Nationwide | Recommendation for home office |
| | Nationwide | Testing of symptomatic cases |
| | Nationwide | Hygiene rules, isolation, quarantine |
| | Cantonal | Restrictions for restaurants and bars |
| (5) January 18, 2021 onwards | Nationwide | Face masks in public transport |
| | Nationwide | Face mask wearing in busy places and public buildings |
| | Nationwide | Ban on gatherings >5 persons |
| | Nationwide | Home office mandatory |
| | Nationwide | Testing of symptomatic cases |
| | Nationwide | Hygiene rules, isolation, quarantine |
| | Nationwide | Closing of restaurants and bars |
| | Cantonal | Vaccination |

**General health and quality of life**: Measured by 1) 1: very poor to poor self-assessed general health status vs 0: very good/good/fair 2) 1: very poor to poor self-assessed quality of life vs 0: very good/good/fair. **Mental health**: Measured by 1) 1: often to always in a depressive mood vs 0: never/seldom/sometimes 2) 1: often to always lacking energy vs 0: never/seldom/sometimes, 3) 1: fear of losing employment vs 0: no fear. **Loneliness/Isolation**: Measured by 1) 1: very often feelings of loneliness vs 0: never/seldom/sometimes/often 2) 1: often feelings of isolation vs 0: never/sometimes (only population 65 years or older). **Physical activity**: Measured by 1) 1: not being physically active vs 0: being physically active. **Health care use**: 1) 1: General health care use vs 0: no use 2) 1: General health care non-use vs 0: no non-use 3) 1: COVID-19 related health care use (contact of general practitioner or hospital because of COVID-19 symptoms) vs 0: no use. **Adherence to mitigation measures**: Measured by 1) 1: always adherence to physical distance when meeting persons vs 0: not always 2) 1: always the wearing of face masks

vs 0: not always 3) 1: always avoidance of private appointments vs 0 not always 4) 1: always non-use of public transport vs 0: not always.

## Variables of interest and confounding variables

Our main variables of interest are language region (German/Romansh, French, Italian) and the above-defined mitigation periods (March 16, 2020 to May 10, 2020; May 11, 2020 to July 5, 2020; July 6, 2020 to October 18, 2020; October 19, 2020 to January 17, 2021; January 18, 2021 onwards). We a priori define the following confounding variables: Age category (<45 years, 45 to <65 years, 65 years or older), gender (women, men), highest attained education (compulsory, secondary, tertiary), nationality (Swiss, non-Swiss), living with a partner (yes/no), living in urban area (yes/no). We selected these variables because we expect an association between the study outcomes and the main variable of interest.

## Statistical methods

We describe the survey population by frequencies (n) and percentages (%). Incidence rates were calculated from Poisson rates with 95% confidence intervals (CIs). We calculate crude proportions of all study outcomes for each period and language region from logistic regression models. We test an interaction effect between language regions and mitigation periods for all study outcomes using a likelihood ratio test (LRT). We test for the null hypothesis of no time trend across mitigation periods by a univariable language region stratified hierarchical logistic regression model (accounting for repeated measurements within participants) using the mitigation periods as independent variable and reporting a two-sided p-value from a LRT [12]. To investigate whether period effects are confounded by other variables, we report odds ratios (OR) and 95% CIs from adjusted language region stratified logistic regression models (i.e. mitigation period as independent variable adjusted for confounding variables). We adjust for the variables *age category*, *gender*, *highest attained education*, *nationality*, *living with a partner* and *living in an urban area*. In hierarchical regression models we scale the calibration weights so that the new weights sum to the effective number of repeated measurements for each participant [13]. We set an alpha level of 5% as statistical significant. We replace missing values by their survey population median values for statistical modeling.

All models are survey-weighted regression models with calibration weights to account for sampling and nonresponse bias and to account for the fact that answers from the same individuals are correlated. Sampling weights make the survey population representative of the Swiss 2018 census population and nonresponse weights account for dropouts and nonresponse. We calculated the probability of being sampled from the census population using a logistic regression model with an age and gender interaction and language region as predictors to construct sampling weights. Non-response weights were constructed in a similar way using predictors *age*, *gender*, *language region*, *living with a partner*, *working situation* and *highest attained education* (see description in S1 Text). All analyses were performed in R version 4.0.2 [14]. For survey-weighted regression, we used the package svyglm version 4.0 [15].

## Ethics statement

Ethical approval: The Cantonal Ethics Commission of Zurich concluded that the current study does not fall within the scope of the Human Research Act (BASEC-Nr. Req-2020-00323).

Informed consent: As per the decision of the Cantonal Ethics Commission of Zurich, explicit informed consent was not needed from participants for this particular study. However, participants gave their general permission to be part of research studies when accepting the

invitation to the online panel from which we sampled our respondents. Participation in the study was voluntary and participants could withdraw from the study at all times.

## Results

### Study population

Fig 1 shows the survey sample distribution by language regions. 2,163 (64%) participants are from the German/Romansh-speaking part of Switzerland, 713 (21%) from the French-speaking part and 505 (15%) from the Italian-speaking part. Table 3 describes the COVID-19 Social Monitor survey population by language region. The total survey sample consists of 3,381 participants. 506 (15.0%) participants are older than 65 years. 48.6% of the survey population are women. Most of the survey participants live in an urban area (80.6%). The average daily COVID-19 incidence per 100,000 inhabitants for the period from February 24, 2020 to December 31, 2020 is 4.24, 95% CI (4.23–4.25), with a substantial variation between language regions. Missing values in baseline characteristics and study outcomes ranged from 0.01% (general health) to 0.4% (education), see S2 Table.

### COVID-19 incidence, by mitigation period and language region

Fig 2 shows new COVID-19 cases per day and 100,000 inhabitants, by mitigation period and language region. The Italian-speaking region had from February 24, 2020 (the first COVID-19

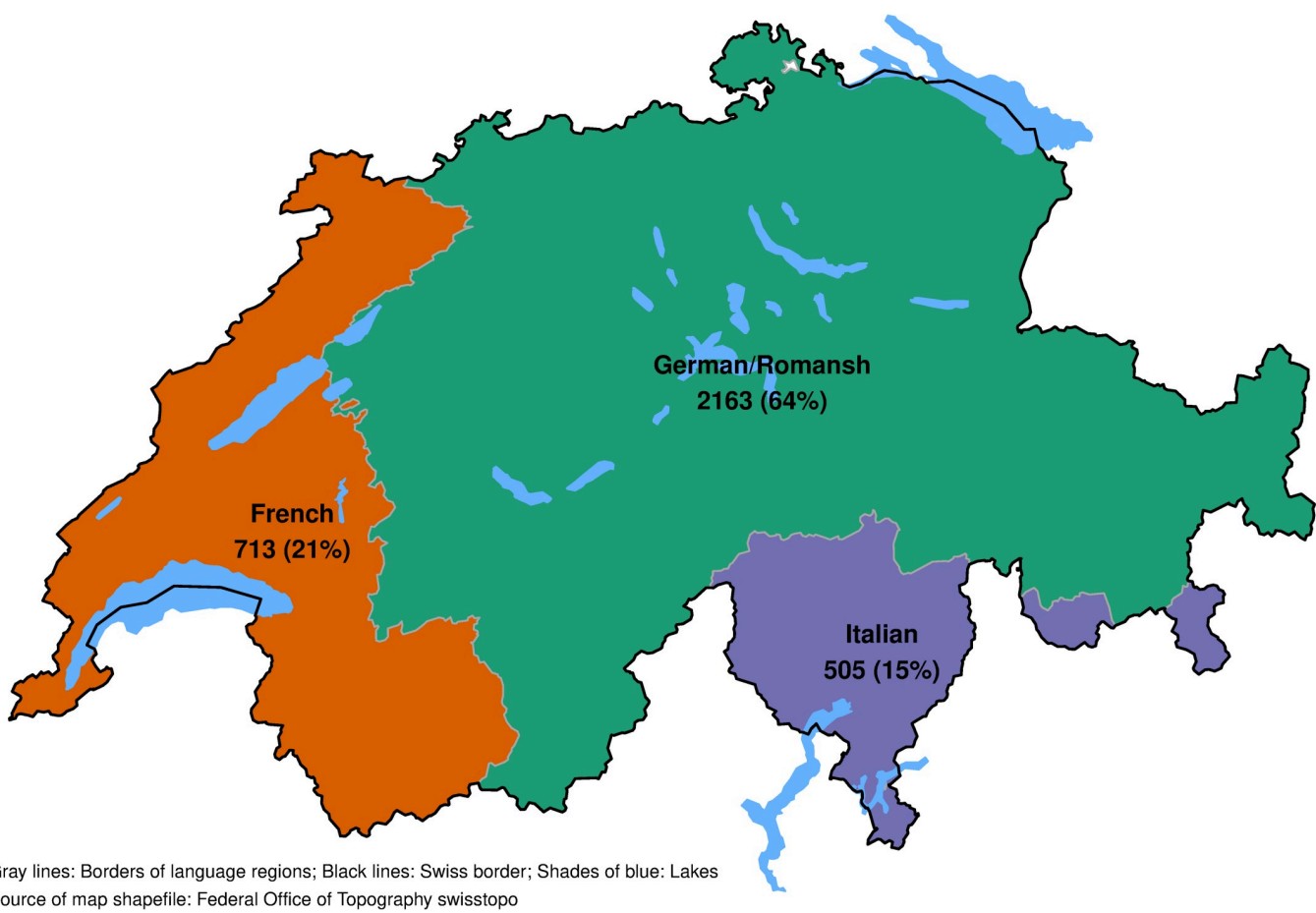

Gray lines: Borders of language regions; Black lines: Swiss border; Shades of blue: Lakes
Source of map shapefile: Federal Office of Topography swisstopo

**Fig 1. Number of survey participants, by language region.**

**Table 3. Survey population baseline characteristics and COVID-19 incidence, by language region.**

| Characteristic | Language region | German/Romansh (n = 2163) | French (n = 713) | Italian (n = 505) | Switzerland (N = 3381) |
|---|---|---|---|---|---|
| | | n (%) | n (%) | n (%) | n (%) |
| Age categories | 0 to <45 years | 1069 (49.4%) | 367 (51.5%) | 253 (50.1%) | 1689 (50.0%) |
| | 45 to <65 years | 765 (35.4%) | 239 (33.5%) | 182 (36.0%) | 1186 (35.1%) |
| | 65 years or older | 329 (15.2%) | 107 (15.0%) | 70 (13.9%) | 506 (15.0%) |
| Gender | Men | 1117 (51.6%) | 359 (50.4%) | 260 (51.5%) | 1736 (51.3%) |
| | Women | 1046 (48.4%) | 354 (49.6%) | 245 (48.5%) | 1645 (48.6%) |
| Highest attained education | Compulsory | 154 (7.1%) | 55 (7.7%) | 23 (4.5%) | 232 (6.9%) |
| | Secondary | 1028 (47.5%) | 340 (47.7%) | 255 (50.5%) | 1623 (48%) |
| | Tertiary | 971 (44.9%) | 316 (44.3%) | 224 (44.4%) | 1511 (44.7%) |
| | missing | 10 (0.5%) | 2 (0.3%) | 3 (0.6%) | 15 (0.4%) |
| Citizenship | Non-Swiss | 150 (7.0%) | 91 (12.8%) | 55 (10.9%) | 296 (8.8%) |
| | Swiss | 2010 (92.9%) | 621 (87.1%) | 449 (88.9%) | 3080 (91.1%) |
| | missing | 3 (0.1%) | 1 (0.1%) | 1 (0.2%) | 5 (0.1%) |
| Working situation | Employed | 1591 (73.6%) | 485 (68%) | 339 (67.1%) | 2415 (71.4%) |
| | Unemployed | 54 (2.5%) | 19 (2.7%) | 19 (3.8%) | 92 (2.7%) |
| | Retired | 328 (15.2%) | 109 (15.3%) | 74 (14.7%) | 511 (15.1%) |
| | Other | 190 (8.8%) | 100 (14.0%) | 73 (14.5%) | 363 (10.7%) |
| Living with partner | Yes | 1527 (70.6%) | 478 (67.0%) | 378 (74.8%) | 2383 (70.5%) |
| Living in urban are | Yes | 1679 (77.6%) | 590 (82.8%) | 455 (90.1%) | 2724 (80.6%) |
| | | Mean (95%CI) | Mean (95%CI) | Mean (95%CI) | Mean (95%CI) |
| COVID-19 incidence per day and 100,000 inhabitants from February 24, 2020 to May 3, 2021 | | 4.25 (4.24–4.26) | 6.72 (6.70–6.6.75) | 5.90 (5.83–5.96) | 4.96 (4.94–4.97) |

case in Switzerland) to May, 10 2020 an average daily incidence of 12.1, 95% CI (11.7, 12.5), per 100,000 inhabitants. The incidence in this region decreased to 0.47, 95% CI (0.38, 0.58), in the second period, increased to 67.9, 95% CI (67.0, 68.9), in the fourth period, and decreased again to 16.9, 95% CI (16.4, 17.3) in the fifth period. We found strong evidence for an interaction effect (p<0.001) between mitigation period and language region.

## Study outcomes

Figs 3–5 show the crude proportions of the study outcomes weighted for sampling and nonresponse by mitigation period, language region and for the whole of Switzerland. For example, the proportion of individuals with no health care use in the German/Romansh-speaking part of Switzerland in the first mitigation period is 14.1%, 95% CI (12.7%-15.7%), and decreases to 1.3%, 95% CI (1.0%-1.8%), in the last mitigation period (see Fig 4 and S3 Table). Individuals from the Italian-speaking part of Switzerland show the highest percentage of adherence to mitigation measures (see Fig 5). We found evidence for a period effect in all language regions for the study outcomes *poor quality of life* (all p<0.04), *depressive mood* (p<0.01), *lack of energy* (all p<0.001), *no physical activity* (all p<0.005), *health care use and non-use* (all p<0.001) and for the *adherence to mitigation measures* (all p<0.001), see S4 Table.

Figs 6–8 show the adjusted ORs from language region stratified hierarchical logistic regression models weighted for sampling and nonresponse. We found evidence for an interaction effect between language region and mitigation period for the study outcome *adherence to mitigation measures* (all p<0.003, see S5 Table). The adjusted OR for not being physically active (compared to the mitigation period March 16, 2020 to May 10, 2020) in the French-speaking part of Switzerland is 1.51, 95% CI (1.16–1.96), in the period from October 19, 2020 to January 17, 2021 (see Fig 6 and S6 Table).

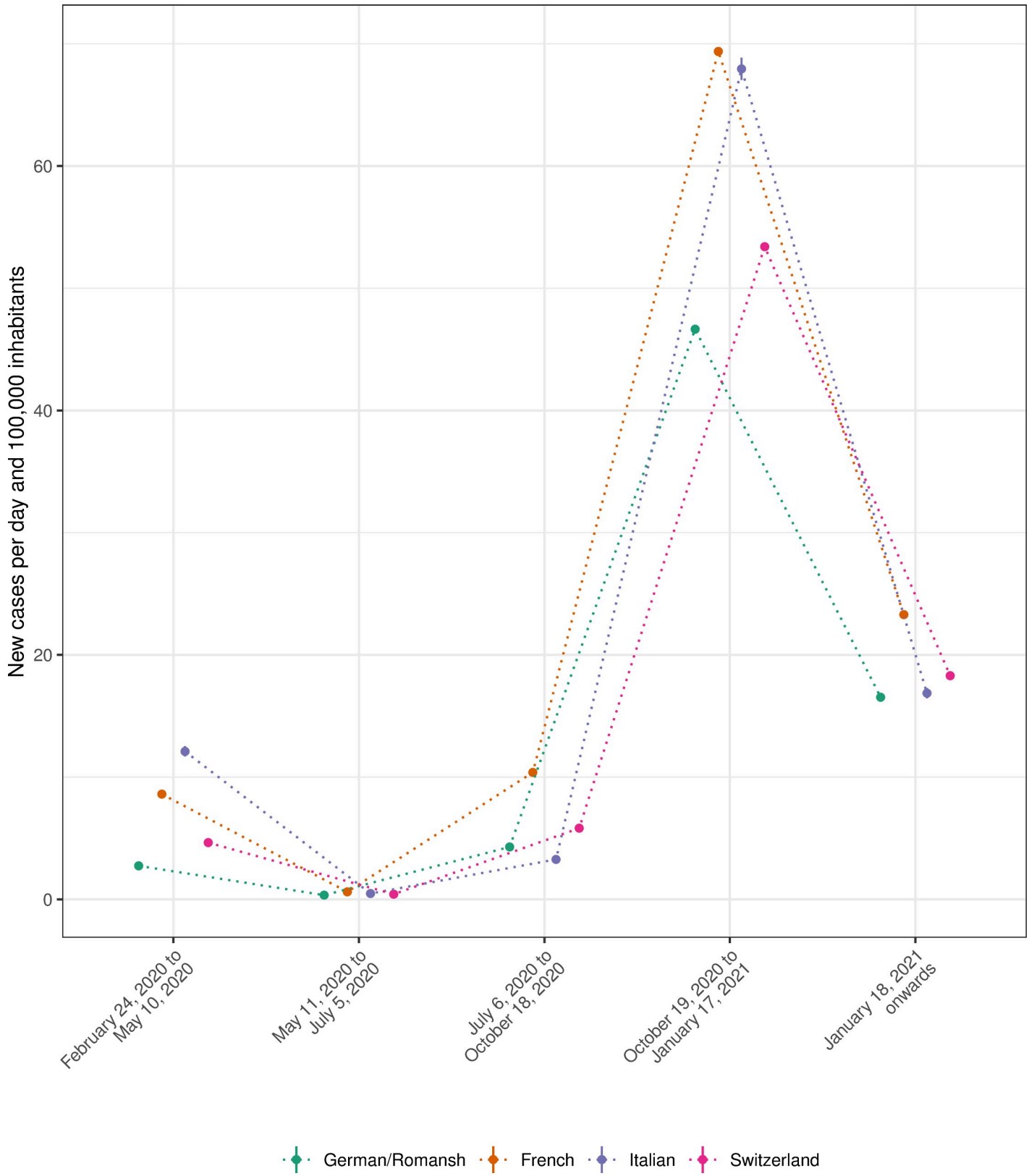

**Fig 2. New COVID-19 cases per day and 100,000 inhabitants, by mitigation period and language region.**

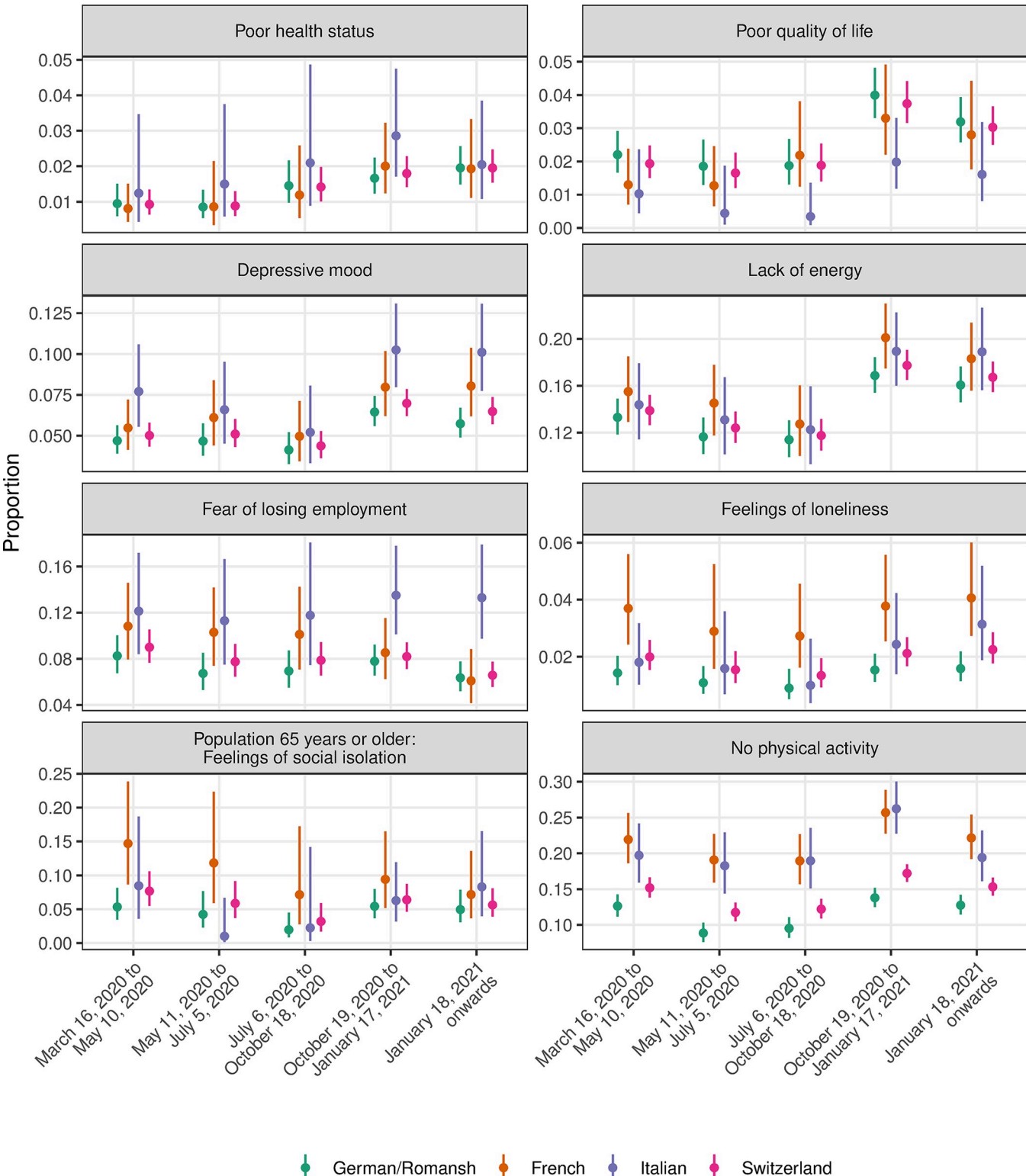

**Fig 3. Proportion of study outcomes for the domains of health, loneliness/isolation and physical activity, by mitigation period, language region and for the whole of Switzerland.**

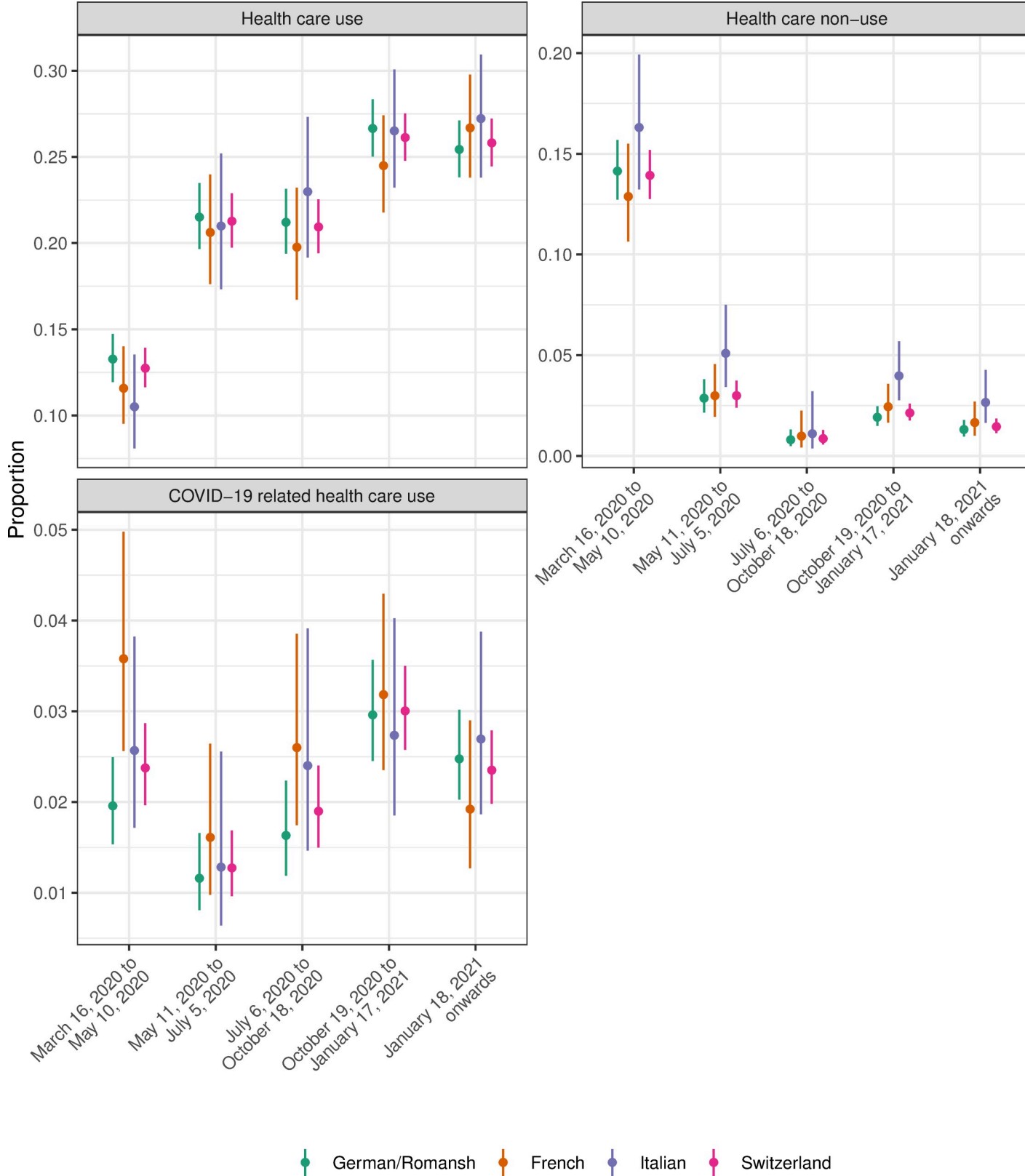

**Fig 4. Proportion of study outcomes for the domain of health care use, by mitigation period, language region and for the whole of Switzerland.**

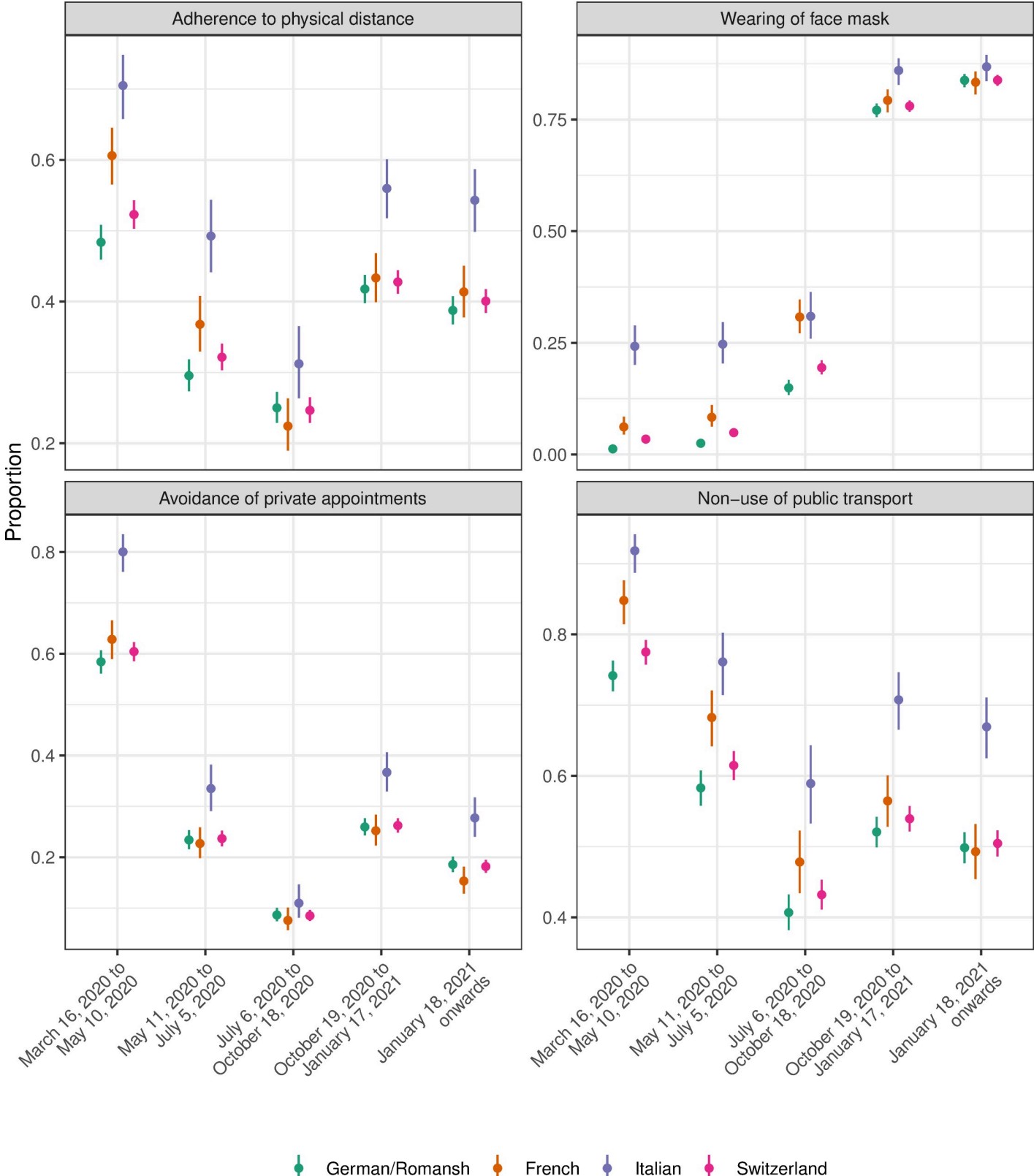

**Fig 5. Proportion of study outcomes for the domain of adherence to mitigation measures, by mitigation period, language region and for the whole of Switzerland.**

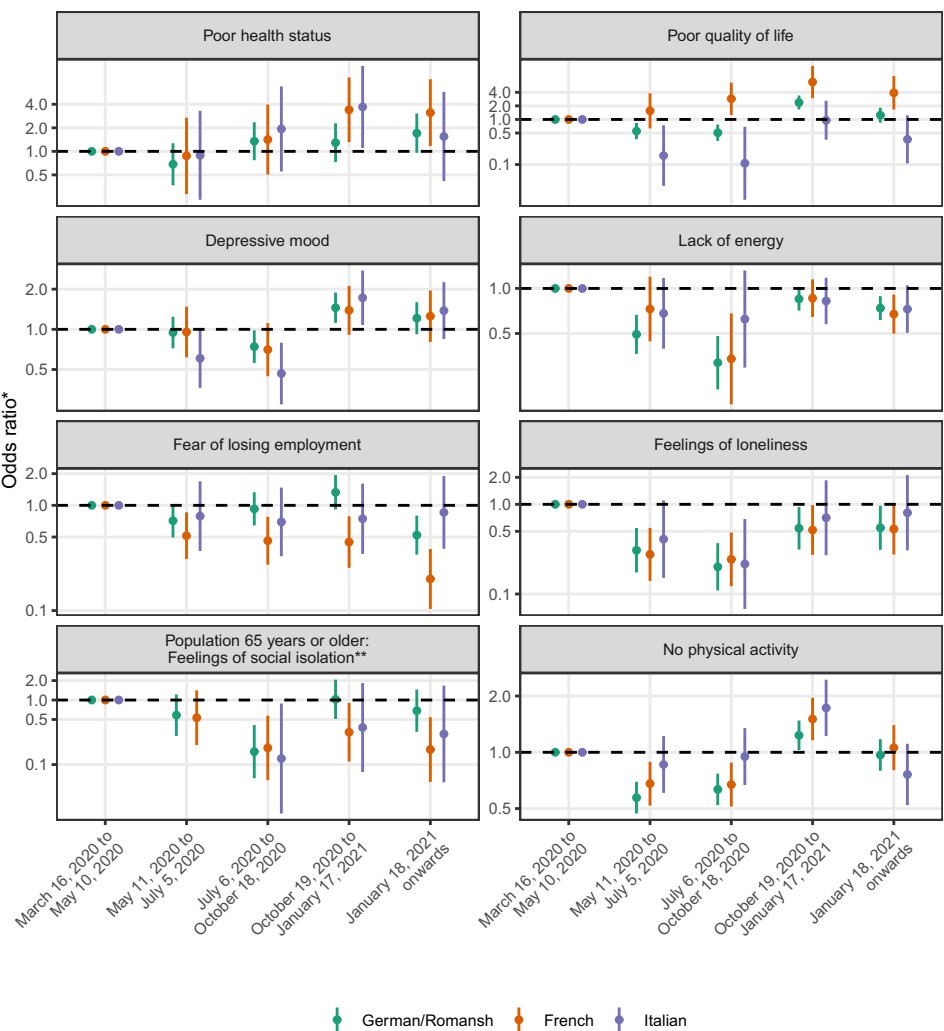

German/Romansh ● French ● Italian

* Reference is mitigation period March 16, 2020 to May 10, 2020.
Adjusted for age, gender, highest attained education, nationality, living with a partner, living in urban area.
** Odds ratio for Italian, May 11, 2020 to July 5, 2020: 0.010, 95% CI (0.0002−0.39).

**Fig 6. Results from adjusted hierarchical logistic regression models for the study outcome domains of health, loneliness/isolation and physical activity, by mitigation period and language region.**

## Discussion

### Summary of main findings

The COVID-19 Social Monitor, a population-based longitudinal online survey, allows us to investigate the impact of mitigation measures on changes in health and social behavior, health care use and the adherence to mitigation measures in Switzerland during the Coronavirus pandemic from March 2020 to December 2020. We hypothesized an interaction effect between mitigation periods and culturally diverse language regions, because of regional and temporal variation in COVID-19 incidence patterns which led to a hetereogeneous implementation of mitigation measures in Switzerland. We found evidence for an interaction effect between language regions and mitigation periods for the study outcome *adherence to mitigation measures*, but not for the other investigated health and social related study outcomes. We observe

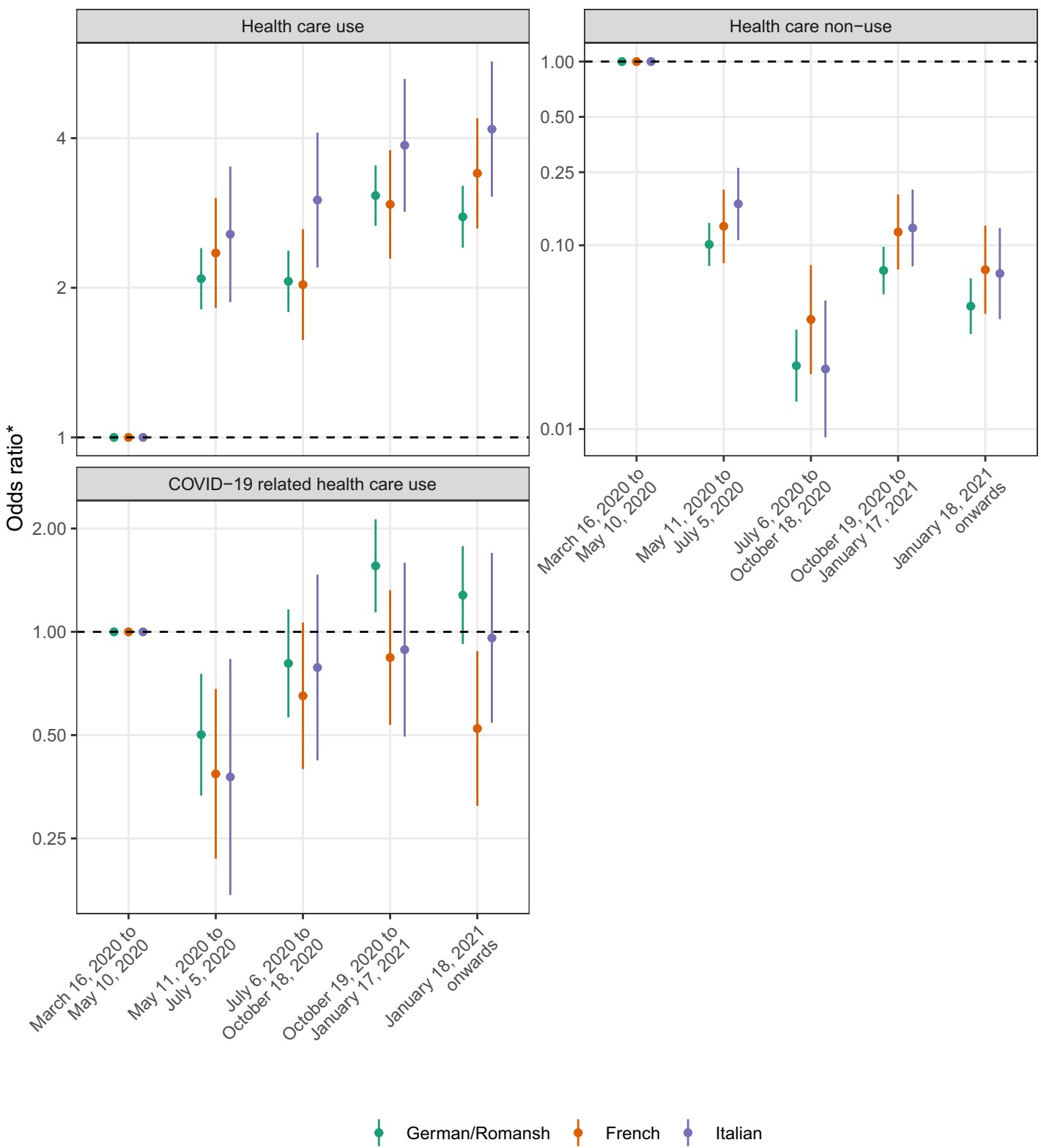

**Fig 7. Results from adjusted hierarchical logistic regression models for the study outcome domain of health care use, by mitigation period, language region and for the whole of Switzerland.**

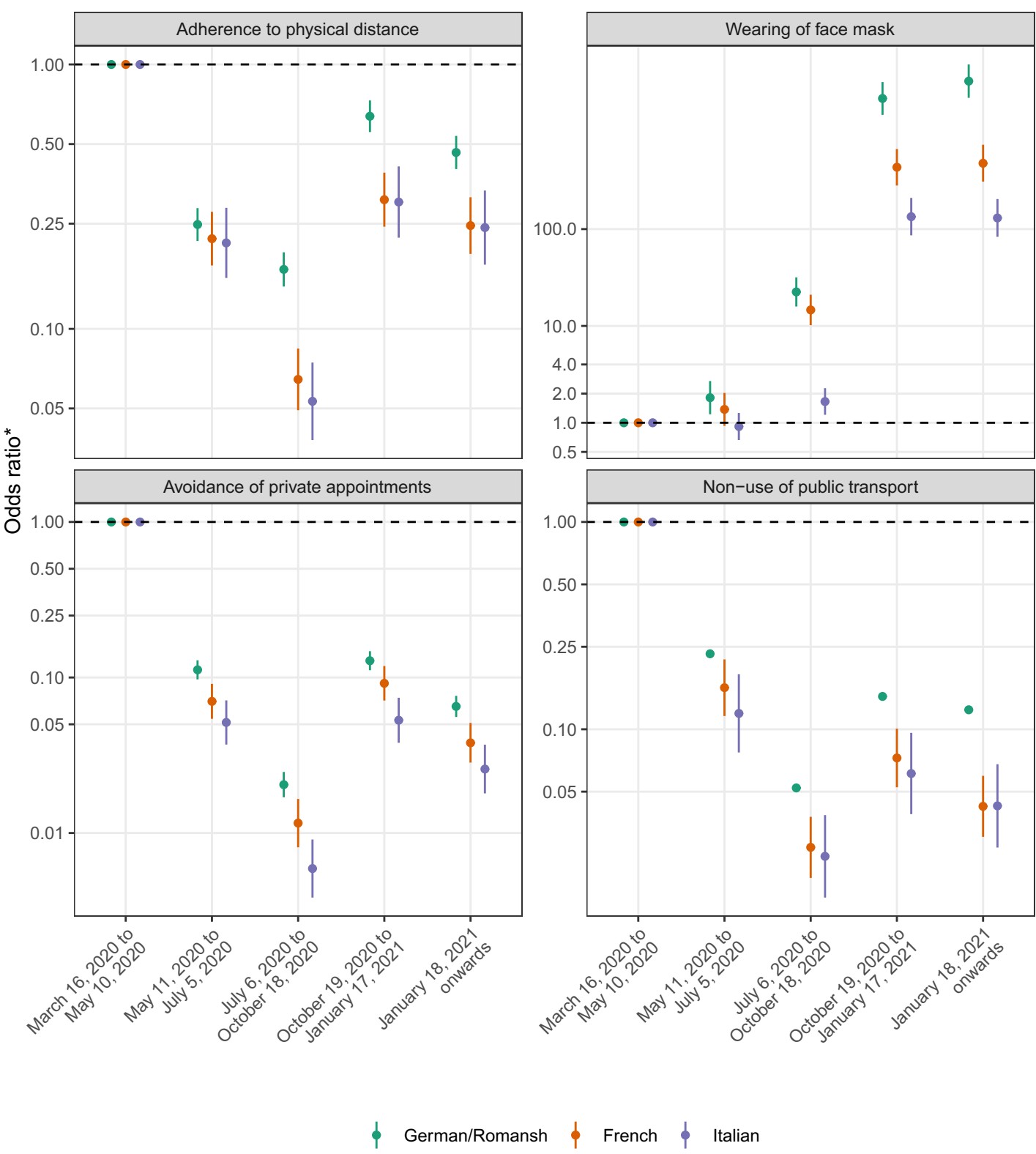

German/Romansh    French    Italian

* Reference is mitigation period March 16, 2020 to May 10, 2020.
Adjusted for age, gender, highest attained education, nationality, living with a partner, living in urban area.

**Fig 8. Results from adjusted hierarchical logistic regression models for the study outcome domain of adherence to mitigation measures, by mitigation period, language region and for whole Switzerland.**

changes in adherence to mitigations measures, with stronger adherence in regions with previously higher COVID-19 incidence. The presence of poor quality of life, depressive mood, lack of energy, no physical activity, general health care use and non-use and the adherence to mitigation measures changed over the analyzed mitigation periods in all language regions. We found no changes in the presence of feelings of loneliness or fear of losing employment over the investigated mitigation periods.

## Regional differences in the course of the epidemic situation

The first COVID-19 case was reported on Febuary 24, 2020, in the canton of Ticino. The canton of Ticino borders the Italian region of Lombardy, which was a highly affected European region in the first wave of the Coronavirus pandemic [16]. The epidemic situation in the canton of Ticino quickly worsened (12.1 daily new cases per 100,000 inhabitants from February 24, 2020 to May 10, 2020) and the cantonal government rapidly implemented stringent mitigation measures to slow down transmission chains. However, new COVID-19 cases quickly appeared in the cantons of Geneva (13.5 new cases 100,000 inhabitants from February 24, 2020 to May 10, 2020), Vaud (9.0 new cases per 100,000 inhabitants from February 24, 2020 to May 10, 2020) and Basel-City (7.5 new cases per 100,000 inhabitants from February 24, 2020 to May 10, 2020) with densely populated areas in and around the larger cities of Geneva, Lausanne and Basel-City. With the nationwide lockdown on March 16, 2020, the incidence rate could quickly be slowed down and stabilized at lower levels. The epidemic situation during the summer months 2020 remained stable at lower incidence rates (0.4 cases per 100,000 inhabitants from May 11, 2020 to July 5, 2020 for the whole of Switzerland) and with a less pronounced variation between regions. In autumn 2020, during the second wave of the Coronavirus pandemic, the incidence rate quickly increases in the French-speaking region (78.7 cases per 100,000 inhabitants from October 19, 2020, onwards), but also in neighbouring regions to Germany (cantons of Basel) and Austria/Italy (cantons Grison, Ticino and Valais). With the worsening situation in the French-speaking region, the cantons of Fribourg, Neuchâtel, Valais and Vaud almost jointly implement more stringent mitigations measures. On October 19, 2020, the Swiss Federal Council announces the mandatory wearing of face masks in public places and buildings and bans gatherings of more than 15 persons. Yet, cantonal authorities react differently for the upcoming Winter months, for example, with varying restrictions for ski resorts. With a stringent implementation of mitigation measures (nationwide closing of restaurants and bars, ban on gatherings with more than 5 people and mandatory homeoffice regulation) from January 18, 2021 onwards, the number of new cases quickly decreased in this period. The Swiss Economic Institute from the Swiss Federal Institute of Technology Zurich estimated that the effective reproduction number decreased in a more pronounced way in cantons with more stringent mitigation measures (https://kof.ethz.ch/en/forecasts-and-indicators/indicators/kof-stringency-index.html, accessed May 20, 2021).

## Changes in health and social behavior

In 2017, the proportion of individuals with a poor self-assessed health status in Switzerland was estimated at 3.5% (German-speaking region: 3.5%, French-speaking region: 3.5%, Italian-speaking region: 5.0%), for feelings of loneliness, at 1.7% (German-speaking region: 1.2%, French-speaking region: 3.0%, Italian-speaking region: 2.5%) and for being physically inactive at 8.2% (German-speaking region: 6.8%, French-speaking region: 10.9%, Italian-speaking region: 13.6%) (https://www.bfs.admin.ch/bfs/de/home/statistiken/gesundheit/erhebungen/sgb.assetdetail.6426300.html). Our survey population estimates for poor self-assessed health status over all investigated mitigation periods are lower than the estimates for 2017, whereas

the estimates for being physically inactive are slightly higher than in 2017. The worsening trend in general health, quality of life and mental health problems from March 2020 to October 2020 onwards may be explained by the strengthening of mitigation measures in October 2020, compared to the very liberal situation during the summer months. This coincides with the increasing incidence rate of COVID-19 in Switzerland. Our findings show that the adherence to mitigation measures quickly changed in regions with higher COVID-19 incidence. The first survey of the COVID-19 Social Monitor by the middle of March 2020 was two weeks after the nationwide lockdown so that citizens were already familiar with the lockdown mitigation measures. The strenghtening mitigation measures from October 2020 may have a large impact on citizens' physical and mental well-being and behaviors. Some of these observed changes may also be influenced by seasonality effects of depressive symptoms and mental health issues [17]. Nevertheless, survey results from Norway and Canada found that stringent mitigation measures are associated with severe mental health problems and with physical inactivity [18, 19]. A systematic review of 68 observational studies of the time period from December 2019 to July 2020 including 19 countries found increased psychological distress which is associated with age, gender, living in rural versus urban areas and socioeconomic position [20]. Population-based cohort studies in Switzerland found that disease outcomes and risk behavior are different across language regions [21–25]. In contrast to other international findings, we could not find a change in the presence of feelings of loneliness during the pandemic [18, 26, 27]. Our survey estimates for feelings of loneliness are similar to estimates from 2017, before the Coronavirus pandemic. Nevertheless, we found changes in the presence of feelings of isolation in the elderly population with a lower chance of feelings of isolation when the mitigation measures were less stringent during the summer months. Social isolation has been shown to be associated with poor health conditions and behavior in the Swiss population [28]. Longterm mental health effects caused by social isolation may be amplified during the Coronavirus pandemic and require further research.

### Health care use during the Coronavirus pandemic

The Coronavirus pandemic has a huge impact on the health care system, including access, delivery and utilisation of health care [29–31]. For example, patients with chronic diseases, acute health events or emergencies may not seek health care during the pandemic with a potential negative impact on longterm health outcomes [32–36]. In general, patients are more likely to be fearful of seeking health care professional advice, non-elective treatments are postponed, and intensive care units in hospitals face an alarming situation with COVID-19 cases. Our results reveal an increased percentage of health care non-use in the first phase of the pandemic with a substantial decrease during the summer months, similar in all language regions. This change may be explained by the improved epidemic situation with less stringent mitigation measures also for health care providers and a seasonality effect during the summer months. Longterm patient outcomes–especially for vulnerable subpopulations and/or the chronically ill–because of a change in health care utilization during the pandemic are still unknown. Regional variation in delivery of health care and health care utilization may be associated with health care (non-)use and may ultimately affect patient outcomes. Switzerland has a substantial variation in health care utilization by region [37–43]. A cross-sectional survey from 2018 in Switzerland found that Italian-speaking individuals reported visiting a specialist more often than individuals living in the French- or German-speaking part of Switzerland [44]. Such regional variation in health care (non-)utilization may have an important impact on population health during the Coronavirus pandemic and needs further investigation, also considering the potential of new telemedicine approaches [45, 46].

## Geographical and socioeconomic factors

Switzerland is a culturally diverse country, surrounded by the countries Austria, France, Germany and Italy. Geographical factors may partly explain some of the regional variation of Coronavirus transmission in Switzerland. Switzerland's mountainous topography divides Southern and Northern Europe and is thus an important European travel link. Italy, Germany and France had a rapid growth in new COVID-19 cases during the first of wave of the pandemic and may reflect the observed incidence patterns in Switzerland [47]. A partial closure of borders to its neighbouring countries aimed to slow transmission rates in Switzerland during the first wave of the pandemic. An important driver for the pandemic is more densely populated areas like cities and surrounding areas leading to an urban-rural gradient [48]. In 2020, the percentage of individuals living in an urban area was estimated at 83% (https://www.bfs.admin.ch/bfs/de/home/statistiken/kataloge-datenbanken/grafiken.assetdetail.12767388.html, accessed January 10, 2020). The COVID-19 incidence patterns in Switzerland show an urban-rural gradient with more reported new cases in cities and urban regions than in rural regions even in pandemic phases with less stringent mitigation measures. Our survey findings show a different adherence to mitigation measures between urban and rural areas with a lower chance of adherence to physical distance, avoidance of private appointments and non-use of public transport for individuals living in urban areas. Such urban-rural behavioral differences may be related to socio-economic factors as socio-economic status substantially varies between regions, cities and even neigbourhoods in Switzerland [49]. Socio-economic factors may play an important role in the adherence to mitigation measures and to social and health behavior during the Coronavirus pandemic [50]. Socio-economic and regional differences in health and social behavior before the pandemic were reported for Switzerland [4, 51, 52]. Our results show that the highest attained education–a proxy for socio-economic position–was associated with changes in health and social behavior in our survey population. For example, our findings show that individuals with tertiary education had a lower chance of being socially isolated, of being physical inactive and of having depressive symtoms compared to individuals with only a compulsory education. Targeted communication strategies may mitigate health inequalities across cultural and socio-economic groups during the pandemic [53]. Nevertheless, further research is needed to investigate health inequalities across socio-economic groups during the Coronavirus pandemic in Switzerland.

## Strengths and limitations

The COVID-19 Social Monitor has several strengths. The longitudinal and population-based survey design allows for a rigorous investigation of behavioral changes during the Coronavirus pandemic. Sampling and nonresponse weights make the survey sample representative of the Swiss 2018 census population older than 18 years. The use of established survey items—which are, for example, also used in the Swiss Health Survey—allows for a comparison of our findings to the year 2017, before the Coronavirus pandemic.

Our study has limitations. First, because our survey is online-based, we probably include more individuals with a greater affinity to online processes and better educated individuals in the survey, which leads to some sampling selection bias [54]. Yet, with the use of sampling weights, we minimize this kind of bias. Second, the regular assessment of individuals with the same survey items may lead to response bias, for example, the tendency to positive answers [55]. Third, we chose mitigation periods based on dates where nationwide mitigation measures were implemented. Federalistically implemented mitigation measures not only varied by canton but also in time. Thus, our chosen mitigation period may not mirror immediate changes

in our investigated study outcomes and may not reflect the true underlying cantonal pattern of those measures. For example, many French-speaking cantons had already implemented a mandatory face mask wearing measure in public stores in early autumn (before October 19, 2020). Fourth, we used dichotomized study outcomes which allows us to present our results in terms of proportions. Thus we have a potential loss of information from categorical variables. Neverthless we think that the benefit from presenting percentages outweighs this limitation.

## Conclusion

We conclude that the implemented mitigation measures from March 2020 to October 2020 had an impact on health and social behavior in Switzerland. The adherence to mitigation measures changed differently between language regions and reflected the COVID-19 incidence patterns in the investigated mitigation periods, with higher adherence in regions with previously higher incidence. Cultural, geographical and socio-economic aspects should be included in future communication strategies and policymaking to diminish potential (and as yet unknown) population health consequences in Switzerland caused by the pandemic. Our study informs the public and health authorities about the positive and negative impacts of implemented mitigation measures on changes in health and social behavior in Switzerland and adds important evidence for public health decision- and policy-making for the targeted implementation of mitigation measures.

## Supporting information

**S1 Table. The domain, the source and the original question from the survey questionnaire.**
(DOCX)

**S2 Table. Description of missing values.**
(DOCX)

**S3 Table. Underlying results for Figs 3–5.**
(DOCX)

**S4 Table. p-values from likelihood ratio test for period effect, by language region.**
(DOCX)

**S5 Table. p-values from likelihood ratio test for interaction effect between language regions and mitigation periods from unadjusted and adjusted models.**
(DOCX)

**S6 Table. Results for from adjusted logistic regression models.**
(DOCX)

**S1 Text. Statistical methods for calibration weights.**
(PDF)

**S2 Text. Reproducible analysis example.**
(PDF)

**S3 Text. Codebook.**
(PDF)

**S1 Data.**
(CSV)

## Acknowledgments

We thank Paul Kelly for proofreading the manuscript.

## Author Contributions

**Conceptualization:** André Moser, Viktor von Wyl, Marc Höglinger.

**Data curation:** Marc Höglinger.

**Formal analysis:** André Moser, Viktor von Wyl, Marc Höglinger.

**Funding acquisition:** Viktor von Wyl, Marc Höglinger.

**Investigation:** André Moser, Viktor von Wyl, Marc Höglinger.

**Methodology:** André Moser, Viktor von Wyl, Marc Höglinger.

**Project administration:** Marc Höglinger.

**Validation:** André Moser, Viktor von Wyl, Marc Höglinger.

**Visualization:** André Moser.

**Writing – original draft:** André Moser, Viktor von Wyl, Marc Höglinger.

**Writing – review & editing:** André Moser, Viktor von Wyl, Marc Höglinger.

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
