## [Decision Letter · Decision Letter 0]

17 May 2021

PONE-D-21-02911

Health and social behaviour through pandemic phases in Switzerland: Longitudinal analysis of the COVID-19 Social Monitor panel study

PLOS ONE

Dear Dr. Höglinger,

Thank you for submitting your manuscript to PLOS ONE. After careful consideration, we feel that it has merit but does not fully meet PLOS ONE’s publication criteria as it currently stands. Therefore, we invite you to submit a revised version of the manuscript that addresses the points raised during the review process.

The paper needs a MAJOR REVISION. Please follow the suggestions given by reviewers in order to improve the quality of the manuscript.

We look forward to receiving your revised manuscript.

Kind regards,

Barbara Guidi

Academic Editor

PLOS ONE

Journal Requirements:

5. Please include a copy of Table 2 which you refer to in your text in line 130.

6.We note that Figure(s) 1 in your submission contain map images which may be copyrighted. All PLOS content is published under the Creative Commons Attribution License (CC BY 4.0), which means that the manuscript, images, and Supporting Information files will be freely available online, and any third party is permitted to access, download, copy, distribute, and use these materials in any way, even commercially, with proper attribution. For these reasons, we cannot publish previously copyrighted maps or satellite images created using proprietary data, such as Google software (Google Maps, Street View, and Earth). For more information, see our copyright guidelines: http://journals.plos.org/plosone/s/licenses-and-copyright.

a)  You may seek permission from the original copyright holder of Figure(s) 1 to publish the content specifically under the CC BY 4.0 license. 

Reviewers' comments:

Reviewer's Responses to Questions

**Comments to the Author**

1. Is the manuscript technically sound, and do the data support the conclusions?

Reviewer #1: No

Reviewer #2: Yes

Reviewer #3: Yes

2. Has the statistical analysis been performed appropriately and rigorously? 

Reviewer #1: No

Reviewer #2: Yes

Reviewer #3: Yes

3. Have the authors made all data underlying the findings in their manuscript fully available?

Reviewer #1: Yes

Reviewer #2: Yes

Reviewer #3: Yes

4. Is the manuscript presented in an intelligible fashion and written in standard English?

Reviewer #1: Yes

Reviewer #2: Yes

Reviewer #3: Yes

5. Review Comments to the Author

Reviewer #1: Thank authors for investigating such an important subject; I have the following comments:

Line 130 states that: “Table 2 shows an overview of four a priori …”, however, table 2 was not included in the paper.

Line143-155 introduces the study outcomes, however, it is important to specify which category is considered as a baseline in each outcome and to report the number and percentages of cases in each category of the outcome.

In Lines 183-185, the authors indicated that they used sampling weights as well as nonresponse weights for the outcomes; although the readers are referred to the supplementary material for detail (I did not have access to supplementary materials), this information is important and should be included in the paper.

Line 181 states that the results are shown by Odds Ratio(OR) with 95%CI; it means that the significant level is set to 0.05, however, in line 262-263, the author reported poor quality of life with a p-value higher than 0.05 as significant.

Line 259-275 explains the interpretation of OR in the figure of 6-8, and it stated that all the variables in figure 6 are significant, however, in these figures most of the confidence intervals include OR=1 so their corresponding outcomes cannot be significant; for example, In line 263 it is stated that “no physical activity” is significant and in line 268 it's OR and 95%CI is reported, however, both ORs’ confidence intervals in line 268 include OR=1 which means that odds ratio for this variable is not significant. As it is demonstrated in figure 6, the variable “no physical activity” is significant only in the last period (0ctober 19, 2020 onwards) for the French and Italian region; similarly, other outcomes in figure 6 have significant value for only the last period.

The main results of the paper including the Odds Ratios and their confidence intervals are shown in figure 6,7, and 8, so the significant variables in the figure needs to be marked so the readers can recognize them.

The title of the paper indicated that longitudinal analysis is performed in the study, however, logistic regression is applied in the current study which is appropriate for cross-sectional study; longitudinal statistical models are the most powerful models for showing the pattern of change during the time, but they are not used in this study.

Reviewer #2: PLOS One review

Thank you for the opportunity to review this manuscript of original study from the Swiss COID19 Social Monitor Study.

My comments follow the order of sections in the manuscript.

Abstract

By reviewing the Abstract it is not really clear what the purpose of the study comprises. Yes, to test predetermined interaction effects in COVID19 mitigation efforts across the three language regions in Switzerland. But why? How do these tests enhance our knowledge in the field of COVID19 or other virus mitigation? Clarifying this would be helpful for readers as many people will only read the abstract.

Introduction

L74-85 and 92-94: This description suggests that mitigation efforts were variant across the 26 cantons more than the three regions that are used in the proposed interaction tests. How will this potentially complicate the storyline, results and interpretation? The statement that variation in mitigation efforts may lead to “different behaviors across language regions” does not appear well supported given this description.

L94-95: “We hypothesize that an interaction effect between pre-specified mitigation periods and language regions on behavioral changes exists”. This appears quite unclear. The authors should provide a cultural description and background information or data which backs the notion that such interaction effects may be warranted. Presently the reader is left in the cold concerning why and which interaction effects may be found between the language regions. Any prospective description around what to except should be outlined in the Introduction section.

Methods

The “Data Source”/Sample subsection is quite challenging to follow. Please state clearly how many individuals were in the original sampling frame, how many responded, and how many have responded in subsequent waves of data collection, including attrition rates.

The subsection defined as “Mitigation periods” appears to suggest that several mitigation periods were indeed applied and led at the federal level. This is in contrast to the Introduction which appears to suggest that most efforts were led at the local level. Please explain.

In line with the comments above about the lack of study question context within the Introduction section, the material presently in the subsection titled “Study outcomes” comes across as very unfocused and lacking in connection with the overall study objective. Specifically, why these measures and not some others are included has not been discussed or contextualized. This is sorely needed.

The description of the measures provided in the “Study outcomes” section is also unclear. Why are all questions dichotomized? Are responses within the same domain collapsed together to form a scale? For example, have the three mental health questions been summed to form a scale or are they modeled individually? Such a description is not provided.

Similar to the above with regards to the confounding variables. Why are those variables and not some others included in the analyses? At this point the reader has not been introduced to any reasoning or context which supports the selection of those variables over other ones.

Analyses

“Because of the low percentage of missing values, we replace missing value by its survey population median value for statistical modeling.” – please reveal the % missing for all waves used on the analyses so that the reader can assess the appropriateness of this method of dealing with missing values.

The analyses subsection is quite chaotic and hard to follow and would require a substantial rewrite for clarity. Presently it is more or less unclear what the analyses entail. For example: “We construct sampling weights to make the survey sample representative of the 2018 census population of Switzerland aged 15 years or older and construct nonresponse weights to account for dropouts and nonresponse (see Supplementary Material for details). We use sampling and nonresponse weights in the above-specified logistic regression models to account for sampling and nonresponse bias.” – by reviewing this statement it is really hard to assess what was actually done and how.

Please clarify if Table 3 includes information about the survey population at baseline only.

It also appears that the authors are conducting trend analyses and interaction effects on time-trends by region. The title of the study should therefore be renamed to include the term “time-trends” as oppose to “longitudinal”. For example, the subtitle cold be “Regional Time-Trends in the COVID19 Social Monitor Panel Study”.

Results

This is very long section that could be truncated.

Discussion

The discussion around variations in mitigation effort further supports the notion that the bulk of variation in observed variables stem from within language region rather than across them.

Reviewer #3: How were participants recruited and followed longitudinally?

Line 299: Is the study truly “population-based” (i.e., people with certain characteristics are more likely to self-select into the survey such as access to information about this study)?

Line 49: For the loneliness/isolation measure, was this only asked among people older than 65 years?

Line 110: It is not clear how many times each participant answered questionnaires multiple times at each wave (followed longitudinally) or if this is a repeated cross-sectional study design? What is the difference between participants from wave 11 and the ‘additionally sampled participants’ from wave 12?

Line 182: Could you quantify the level of missingness?

Was multiple testing addressed?

304: typo “varyiance”

Line 318: What is the purpose of this section “The course of the epidemic situation in Switzerland?” It does not refer to the data/results and seems out of place.

What are some of the limitations that come with having a dichotomous yes/no for all the outcome variables?

6. PLOS authors have the option to publish the peer review history of their article (what does this mean?). If published, this will include your full peer review and any attached files.

Reviewer #1: No

Reviewer #2: No

Reviewer #3: No

---

## [Author Response · Author response to Decision Letter 0]

4 Jun 2021

Our responses to the reviewers and the editor might be more easily readable in the separate letter instead of in this form.

We checked the PLOS One style requirements and corrected formatting and changed the file naming accordingly. Also, we included the required references to the supplementary files at the end of the article.

We now offer more details regarding participant consent in the manuscript:

Informed consent: As per the decision of the Cantonal Ethics Commission of Zurich, explicit informed consent was not needed from participants for this particular study. However, participants gave their general permission to be part of research studies when accepting the invitation to the online panel from which we sampled our respondents. Participation in the study was voluntary and participants could withdraw from the study at all times.

Zurich University of Applied Sciences, to which the corresponding author is affiliated, is a PLOS institutional partner with direct billing option. We have used this in the past; hence, that should work.

We included captions in the Supporting Information (Tables in S1-S5 Table) and changed in-text citations accordingly.

5. Please include a copy of Table 2 which you refer to in your text in line 130.

We apologize for the missing Table 2. This table is now included in the manuscript.

6.We note that Figure(s) 1 in your submission contain map images which may be copyrighted. All PLOS content is published under the Creative Commons Attribution License (CC BY 4.0), which means that the manuscript, images, and Supporting Information files will be freely available online, and any third party is permitted to access, download, copy, distribute, and use these materials in any way, even commercially, with proper attribution. For these reasons, we cannot publish previously copyrighted maps or satellite images created using proprietary data, such as Google software (Google Maps, Street View, and Earth). For more information, see our copyright guidelines: http://journals.plos.org/plosone/s/licenses-and-copyright.

We used shapefiles from the Federal Office of Topography (swisstopo) for plotting the maps. The free geodata and geoservices of swisstopo may be used, distributed and made accessible. (https://www.swisstopo.admin.ch/en/home/meta/conditions/geodata/ogd.html).

5. Review Comments to the Author

Reviewer #1: 

We thank reviewer #1 for her or his feedback on our manuscript. We hope to clarify in our answers below that we did not a priori define a “significant” type-I error cutoff to avoid a dichotomization of findings. We would also like to emphasize that we did not use the wording “significant” in the manuscript. Instead, we utilize descriptors for different levels of ‘strength of evidence’, which allow readers to judge by themselves how ‘strong’ this evidence is. Not only is this approach recommended by eminent statisticians and reporting guidelines in our field (e.g. STROBE), we also believe that this is an appropriate way to discuss our findings, because under our observational study design - which is likely influenced by sampling, non-response and selection bias – any effect estimate needs a careful interpretation, which should not be influenced by p-values. P-values were only reported in two cases (due to the lack of better alternatives): for interaction tests (S6 Table) and in an overall test for period effects (S5 Table).

Thank authors for investigating such an important subject; I have the following comments:

Line 130 states that: “Table 2 shows an overview of four a priori …”, however, table 2 was not included in the paper.

We apologize for the missing Table 2. This table is now included in the manuscript.

Line 143-155 introduces the study outcomes, however, it is important to specify which category is considered as a baseline in each outcome and to report the number and percentages of cases in each category of the outcome.

Thank you for this important comment. We added the corresponding information (1 vs 0) in the methods section. The percentages of the study outcomes are shown in Figures 3-5 and absolute numbers are provided in S3 Table.

In Lines 183-185, the authors indicated that they used sampling weights as well as nonresponse weights for the outcomes; although the readers are referred to the supplementary material for detail (I did not have access to supplementary materials), this information is important and should be included in the paper.

We apologize that the reviewer did not have access to the supplementary material, which was uploaded during the submission process. We agree that it is a very important piece of information for the reader. However, the corresponding section in the supplementary material is likely too technical for non-statistical readers. We decided to include a more detailed description of our weighting strategy in the methods section and then still refer to the supplementary material section.

Line 181 states that the results are shown by Odds Ratio(OR) with 95%CI; it means that the significant level is set to 0.05, however, in line 262-263, the author reported poor quality of life with a p-value higher than 0.05 as significant.

We thank the reviewer for this thoughtful comment. We would like to stress that we decided not to a priori define a pre-specified cut-off type-I error of say, 5% (we additionally note that the manuscript does not include the wording “significant”). We based this decision on the ASA Statement on p-Values (https://doi.org/10.1080/00031305.2016.1154108), the observational design of our study and all the required assumptions on our used statistical models. We do not discuss “significant” results but discuss whether evidence for an association exists. Thus, we leave it up to the reader whether a p-value of, say 0.1, is enough “evidence” (under the assumptions that the null hypothesis is true, the statistical models are true and the frequentist properties of the p-value). We agree with the reviewer that our reported list of “p-values” is arbitrary, but we wanted to be parsimonious in the results section (and not mention all estimates). We agree with the reviewer that the 95% width choice of the confidence interval is arbitrary. However, we had to define a range to show the uncertainty of our estimates. We think that a 95% CI is an appropriate choice for presenting uncertainty around point estimates. But we would like to emphasize that we did not intend this range to be used for dichotomizing results into “significant” or “not significant” results.

Line 259-275 explains the interpretation of OR in the figure of 6-8, and it stated that all the variables in figure 6 are significant, however, in these figures most of the confidence intervals include OR=1 so their corresponding outcomes cannot be significant; for example, In line 263 it is stated that “no physical activity” is significant and in line 268 it's OR and 95%CI is reported, however, both ORs’ confidence intervals in line 268 include OR=1 which means that odds ratio for this variable is not significant. As it is demonstrated in figure 6, the variable “no physical activity” is significant only in the last period (0ctober 19, 2020 onwards) for the French and Italian region; similarly, other outcomes in figure 6 have significant value for only the last period.

We thank the reviewer for this comment. We relate our answer to the previously- addressed revision point that we did not use a dichotomizing cut-off type-I error of 5%. We stress that we do not mention “significant” results on lines 259-275. We discuss whether evidence for an association exists and leave it up to the reader to decide how strong this evidence is. We agree that our chosen way of talking about evidence is one of many options, but it is a preferred and recommended way in epidemiological research to present results (see e.g. ISBN 978-0-865-42871-3). Of course, p-values contradict in general the likelihood principle so that the terminology of “evidence” requires a careful use. Given that the likelihood function covers all necessary information for parameter inference, other measures should be reported for completeness (such as likelihood ratios or relative likelihood ratios, see, for example, 10.1007/978-3-662-60792-3). However, we decided not to include additional measures to avoid confusion for the reader. We agree with the reviewer that the presented 95% CIs is arbitrary, but we emphasise again that we had to decide for a range of uncertainty for presenting results. 

The main results of the paper including the Odds Ratios and their confidence intervals are shown in figure 6,7, and 8, so the significant variables in the figure needs to be marked so the readers can recognize them.

We thank the reviewer for this comment. We decided not to dichotomize results into “significant” or “not significant”. Given that our study design is a survey design (thus, of observational nature) marking “significant” results (as the reviewer suggests) is misleading to the reader. Observational designs require the careful inclusion of observed (and unobserved) confounders to make estimates as “unbiased” (in terms of removing spurious association from the design and sampling process) as possible. However, this is only possible to some degree, so that marking “significant” results is not recommended. We assume that potential readers of our manuscript have basic knowledge of epidemiological, biostatistical and causal approaches, and are therefore in a position to assess the strength of evidence of our findings (without dichotomizing by “significance”). Hence, we preferred not to implement the suggestion by the reviewer.

The title of the paper indicated that longitudinal analysis is performed in the study, however, logistic regression is applied in the current study which is appropriate for cross-sectional study; longitudinal statistical models are the most powerful models for showing the pattern of change during the time, but they are not used in this study.

Thank you for bringing up this important point, which has made us realize that further explanations of our regression approach are warranted. Logistic regression models “model” the likelihood function of the statistical model - which is in our case a Bernoulli distributed random variable - which transforms – in case ofidentically and independently distributed random variables - into a binomial likelihood. This is independent, whether measurements from the same individuals are correlated or not. Thus, a logistic regression model can be used for nested (and unnested) designs. Our study design requires the inclusion of calibration weights (which accounts for the sampling design and non-response), but also that (repeated) measurements from the same individual are not independent. Thus, we used a survey-design approach (modeled with the R package svyglm). In brief, this model approach accounts for the sampling design and non-response, while accounting for the fact that measurements are nested within participants (this is specified in the analysis as svyglm(id~id, …), where id is the variable for identifying participants. We modeled the period effect as a fixed effect because those effects are common for all participants and in order to investigate our a priori specified interaction effect between language regions and mitigation periods. Thus, we perform a “longitudinal” analysis using logistic regression models. We agree with the reviewer that we were not precise in how we included the fact that measurements of the same individuals are correlated. We added a sentence to the methods section for clarification and added a reference for the used statistical package.

Reviewer #2: PLOS One review

We are grateful for the feedback and comments of reviewer #2, which greatly improved the readability of our manuscript. We agree that in specific sections we were too vague and ideas were not precisely formulated. We hope that this revised version better explains our aims and storyline of the manuscript.

Thank you for the opportunity to review this manuscript of original study from the Swiss COID19 Social Monitor Study.

My comments follow the order of sections in the manuscript.

Abstract

By reviewing the Abstract it is not really clear what the purpose of the study comprises. Yes, to test predetermined interaction effects in COVID19 mitigation efforts across the three language regions in Switzerland. But why? How do these tests enhance our knowledge in the field of COVID19 or other virus mitigation? Clarifying this would be helpful for readers as many people will only read the abstract.

We thank the reviewer for this helpful comment. We added a sentence to the background that our findings aim to support targeted implementation of mitigation measures, while accounting for cultural aspects. We believe that these finding are relevant not only for Switzerland but for Europe in general.

Introduction

L74-85 and 92-94: This description suggests that mitigation efforts were variant across the 26 cantons more than the three regions that are used in the proposed interaction tests. How will this potentially complicate the storyline, results and interpretation? The statement that variation in mitigation efforts may lead to “different behaviors across language regions” does not appear well supported given this description.

We thank the reviewer for this important comment. It is true that – due the federal system in Switzerland – the cantonal organization play an important role in the variation of mitigation measures. However, language regions play a very important role in health and social behavior in Switzerland. This is supported by the cited literature in the discussion with evidence from large nationwide cohort studies about health and social aspects. We do not think that the nested structure of cantons within language regions complicate the storyline and our findings. Cultural behavior and tradition play a very important role in Switzerland, such that the investigation of a ‘higher-level factor’ language region is not only well-established in the literature (see e.g. references 4, 7, 8, 20-23 in the revised manuscript), but also necessary. In the discussion we highlighted how the pandemic emerged geographically, which was likely influenced – among other factors - by neighboring countries. Our findings show that the Italian-speaking part of Switzerland still has a different behavior in relation to mitigation measures than the other parts of Switzerland. We think that mostly overlaps with the cultural aspects which ultimately influence individual behavior. 

L94-95: “We hypothesize that an interaction effect between pre-specified mitigation periods and language regions on behavioral changes exists”. This appears quite unclear. The authors should provide a cultural description and background information or data which backs the notion that such interaction effects may be warranted. Presently the reader is left in the cold concerning why and which interaction effects may be found between the language regions. Any prospective description around what to except should be outlined in the Introduction section.

We thank the reviewer for this important comment. We changed the introduction accordingly. We now give an example (vaccination uptake before the pandemic) and mention the differences in administered COVID-19 vaccination rates across language regions. Further, we now explicitly mention the influence of Switzerland’s surrounding countries and the coping strategies which vary over the pandemic phases and language regions, leading to our hypothesized interaction effect. We hope this clarifies our interaction hypothesis for the reader.

Methods

The “Data Source”/Sample subsection is quite challenging to follow. Please state clearly how many individuals were in the original sampling frame, how many responded, and how many have responded in subsequent waves of data collection, including attrition rates.

We rewrote this section accordingly. Additionally, we changed Table 1, which now shows all used survey waves with the number of participants and attrition.

The subsection defined as “Mitigation periods” appears to suggest that several mitigation periods were indeed applied and led at the federal level. This is in contrast to the Introduction which appears to suggest that most efforts were led at the local level. Please explain.

We thank the reviewer for this comment. We agree that Table 2 likely provides a picture that cantons only have a minor role in the implementation of mitigation measures. We want to clarify as follows: 1) Some of the mitigation measures include the (repetitive) ‘standard’ measures of hygiene rules, isolation, testing, face masks and others. Those measures have been implemented during the very early phase of the pandemic on a federal level. Thus, Table 2 has many repetitive entries on a “nationwide” level, which likely gives the impression that federal measures have higher priority. The Swiss Federal Council announced the state of emergency from March 16, 2020, to June 19, 2020. From then on, cantons were responsible for the implemented mitigation measures until January 18, 2021 (measures for restaurants, bars, face mask wearing). Those cantonal mitigation measures had a major impact on individual behavior and mobility and ultimately on the emerging of new waves. For example, during the autumn months of 2020, individuals traveled from a more stringent canton to a less stringent canton for, for example, shopping and leisure activities, which was likely a driver for increased new cases. Thus, the cantonal mitigation played a huge role in the development of the pandemic for a long period. However, we still think that Table 2 should include all measures as listed, because all play an important role in Switzerland’s mitigation strategy.

In line with the comments above about the lack of study question context within the Introduction section, the material presently in the subsection titled “Study outcomes” comes across as very unfocused and lacking in connection with the overall study objective. Specifically, why these measures and not some others are included has not been discussed or contextualized. This is sorely needed.

We agree that our reasoning on the selection of study outcomes is vague. We a priori selected study outcomes to cover a broad domain of relevant health and behavioral aspects, while accounting for the necessity that study outcomes need to be included in all survey wave questionnaires (this is not the case for all study outcomes in our study). We changed this section accordingly. Also, we now mention in the introduction (objectives subsection) that our aim is to analyze in a descriptive way variations in relevant health and behavioral outcomes over the course of the pandemic and corresponding differences across language regions. 

The description of the measures provided in the “Study outcomes” section is also unclear. Why are all questions dichotomized? Are responses within the same domain collapsed together to form a scale? For example, have the three mental health questions been summed to form a scale or are they modeled individually? Such a description is not provided.

We dichotomized study outcomes so that results can be communicated in terms of proportions and odds ratios. We are aware that we lose information through this process. However, we think the benefit for the communication of results in proportions outweighs the potential information loss due to the dichotomization. The same comment has been pointed out by reviewer #3. 

Similar to the above with regards to the confounding variables. Why are those variables and not some others included in the analyses? At this point the reader has not been introduced to any reasoning or context which supports the selection of those variables over other ones.

We apologize for being imprecise in the formulations. We now explicitly mention that we expect an association of those variables with the study outcomes and language region. Those variables are known confounders and have been used in many epidemiological studies (see e.g. references 20-23, 47, 50 in revised manuscript).

Analyses

“Because of the low percentage of missing values, we replace missing value by its survey population median value for statistical modeling.” – please reveal the % missing for all waves used on the analyses so that the reader can assess the appropriateness of this method of dealing with missing values.

We thank the reviewer for this important comment. We included a Supplemental Table which presents the number of missing values for baseline characteristics and study outcomes. Because of the low frequency, we decided to show only the overall missing number and not per wave. We expect a very small bias from the median imputation.

The analyses subsection is quite chaotic and hard to follow and would require a substantial rewrite for clarity. Presently it is more or less unclear what the analyses entail. For example: “We construct sampling weights to make the survey sample representative of the 2018 census population of Switzerland aged 15 years or older and construct nonresponse weights to account for dropouts and nonresponse (see Supplementary Material for details). We use sampling and nonresponse weights in the above-specified logistic regression models to account for sampling and nonresponse bias.” – by reviewing this statement it is really hard to assess what was actually done and how.

Thank you for this suggestion to make it less ‘chaotic’. Nevertheless, we are a bit unclear as to what it makes so chaotic, because the section is structured in a rather common way: Used descriptive statistics, used modeling approaches (including used test statistics, effect measures and adjustment variables) and dealing with missing values. This is a common and suggested way to structure the statistical analysis section. We rewrote the calibration weights part with more focus on why we use those weights and set it apart in a new paragraph. We hope this clarifies the used statistical approaches and simplifies it for the reader . Some rewriting is connected to comments from reviewer #1 who required a better description of the calibration weights.

Please clarify if Table 3 includes information about the survey population at baseline only.

Thank you for this suggestion. We added that we only show baseline characteristics.

It also appears that the authors are conducting trend analyses and interaction effects on time-trends by region. The title of the study should therefore be renamed to include the term “time-trends” as oppose to “longitudinal”. For example, the subtitle cold be “Regional Time-Trends in the COVID19 Social Monitor Panel Study”.

Thank you for this suggestion. We changed to this more informative subtitle.

Results

This is very long section that could be truncated.

We substantially shortened the results section. We decided to delete the supporting information about adjusted effects of education and urban/rural information, because this is only briefly discussed in the discussion but not as a main finding. We decided on the other hand to provide the estimates for Figures 3-6, and also of the statistical tests for interaction and period effects.

Discussion

The discussion around variations in mitigation effort further supports the notion that the bulk of variation in observed variables stem from within language region rather than across them.

Thank you for this comment. We agree that due to the federal structure in Switzerland some of the variation might be explained by cantons. However, language regions play an important overall role in social and health behavior, structural organizational and even political decisions. Several studies from Switzerland show evidence for strong cultural differences (e.g. in risk behavior or vaccination uptake, see references 7 and 8 in the manuscript). Working and leisure time mobility of Swiss residents is often within the same language region, which can be seen in defined labor force areas (https://www.bfs.admin.ch/bfs/de/home/grundlagen/raumgliederungen.assetdetail.8706492.html). This supports the hypothesis that cultural regions have a higher-level impact on every-day life activities of Swiss residents (and thus the mitigation or emerging of the pandemic), despite the federal system in Switzerland. The Swiss Economic Institute of the Swiss Federal Institute of Technology in Zürich summarized that the effective reproduction number decreased in a more pronounced way in cantons with stronger mitigation measures (https://kof.ethz.ch/en/forecasts-and-indicators/indicators/kof-stringency-index.html). Given that the strengthening of mitigation measures in the autumn months of 2020 were often joint efforts from cantons within the same language region, this also supports a cultural effect of mitigation measures. We conclude that we can only speculate as to whether differences in mitigation adherence across language regions might be in truth more pronounced than our survey results indicate.

Reviewer #3: 

We greatly appreciate the comments of reviewer #3, which clarify aspects of important open study design and reporting points. Thank you!

How were participants recruited and followed longitudinally?

Line 299: Is the study truly “population-based” (i.e., people with certain characteristics are more likely to self-select into the survey such as access to information about this study)?

We thank the reviewer for this comment and have added a sentence on the characteristics of our sample. Yes, it is ‘population-based’, because the study design allows the generalizability of our findings to the Swiss population. Study participants have been sampled from an online-panel whose members have been actively recruited using random probability sampling based on national landline telephone directories and random digit dialing of mobile phone numbers. Of course, response probability as well as sample attrition might be correlated to personal characteristics such as online-affinity. Even if the survey sample is prone to selection bias, the sampling process was defined in the whole Swiss population and with the construction of calibration weights we address potential biases which would affect the generalizability of our findings.

Line 49: For the loneliness/isolation measure, was this only asked among people older than 65 years?

Yes, this question was only asked among individuals older than 65 years.

Line 110: It is not clear how many times each participant answered questionnaires multiple times at each wave (followed longitudinally) or if this is a repeated cross-sectional study design? What is the difference between participants from wave 11 and the ‘additionally sampled participants’ from wave 12?

The initial sample from March 2020 consisted of approximately 2,000 participants. Although retention was overall very good, some individuals dropped out during the course of the first 11 waves. Therefore, we decided to replenish the sample while maintaining representativeness. The new participants included in wave 12 were recruited from the same online panel using the same sampling strategy.

Line 182: Could you quantify the level of missingness?

Was multiple testing addressed?

We thank the reviewer for this important question. Our study aim was to investigate a (joint) period effect across different regions in Switzerland. Thus, we aim to investigate a (stratified by region) null hypothesis of no period effect for different study outcomes. However, as stated in our answers to the comments of reviewer #1, we decided not to specify an a priori type-I error of, say 5%. We believe that this a misleading approach given our study design. Given that a multiple comparison adjustment corrects a p-value conditioning under the null hypothesis, this ‘adjusted’ p-value is only valid under the null hypothesis given the true survey sampling mechanism and statistical model. This can of course be justified in a more ‘controlled’ setting (‘best’ example would be a randomized controlled trial with a priori defined interim analysis), but which was not possible for this survey design. Even under a multiple comparison adjustment, we would report our results in the same way, independent of whether an ‘adjusted’ p-value is below or above a certain (adjusted) threshold.

304: typo “varyiance”

Thank you for spotting this typo.

Line 318: What is the purpose of this section “The course of the epidemic situation in Switzerland?” It does not refer to the data/results and seems out of place.

We aimed to describe where the pandemic in Switzerland emerged (Ticino) and progressed (to areas of Zurich and Basel) and how quickly. We thought that this will lead to a better understanding of the incidence pattern in Switzerland, and also where cantonal mitigation measures were selectively implemented over time. We believe that this might explain some of our findings, because it includes important cultural aspects. We reformulated the title of this section to make our intention more explicit: Regional differences in the course of the epidemic.

What are some of the limitations that come with having a dichotomous yes/no for all the outcome variable

A potential limitation is the loss of information from categorical variables. Despite choosing ‘suitable’ cut-offs (for example, combining very poor and poor self-assessed health status), we lose relevant information from the other categories. On the other hand, dichotomization allows us to express estimates in term of percentages, which is more easily understandable than estimates from a Likert-scale. We think that the loss of information is justified by the gain in clarity. 

We thank the Editor and the reviewers for their helpful feedback which substantially improved our manuscript.

Marc Höglinger (and Co-authors)

---

## [Decision Letter · Decision Letter 1]

6 Jul 2021

PONE-D-21-02911R1

Health and social behaviour through pandemic phases in Switzerland: Regional time-trends of the COVID-19 Social Monitor panel study

PLOS ONE

Dear Dr. Höglinger,

Thank you for submitting your manuscript to PLOS ONE. After careful consideration, we feel that it has merit but does not fully meet PLOS ONE’s publication criteria as it currently stands. Therefore, we invite you to submit a revised version of the manuscript that addresses the points raised during the review process.

ACADEMIC EDITOR: Please insert comments here and delete this placeholder text when finished. Be sure to:

Indicate which changes you require for acceptance versus which changes you recommendAddress any conflicts between the reviews so that it's clear which advice the authors should followProvide specific feedback from your evaluation of the manuscript

We look forward to receiving your revised manuscript.

Kind regards,

Lucinda Shen

Staff Editor 

on behalf of

Barbara Guidi

Academic Editor

PLOS ONE

Journal Requirements:

Additional Editor Comments (if provided):

Reviewers' comments:

Reviewer's Responses to Questions

**Comments to the Author**

1. If the authors have adequately addressed your comments raised in a previous round of review and you feel that this manuscript is now acceptable for publication, you may indicate that here to bypass the “Comments to the Author” section, enter your conflict of interest statement in the “Confidential to Editor” section, and submit your "Accept" recommendation.

Reviewer #1: (No Response)

Reviewer #2: All comments have been addressed

Reviewer #3: All comments have been addressed

2. Is the manuscript technically sound, and do the data support the conclusions?

Reviewer #1: No

Reviewer #2: Yes

Reviewer #3: Yes

3. Has the statistical analysis been performed appropriately and rigorously? 

Reviewer #1: No

Reviewer #2: Yes

Reviewer #3: Yes

4. Have the authors made all data underlying the findings in their manuscript fully available?

Reviewer #1: Yes

Reviewer #2: Yes

Reviewer #3: Yes

5. Is the manuscript presented in an intelligible fashion and written in standard English?

Reviewer #1: Yes

Reviewer #2: Yes

Reviewer #3: Yes

6. Review Comments to the Author

Reviewer #1: The paper has two main problems that the authors did not address:

First, the statistical method applied inappropriately; second, the statistical interpretation of the results is incorrect.

1- For investigating the pattern of change during the time, the statistical model must have the ability to investigate the change of outcome during the time, but the logistic regression that was applied in this study is only able to estimate the outcome (odds ratio in this paper) for the specific time point and cannot compare the odds ratios in different time points to show whether the pattern of change exists or not; for showing the pattern of change during the time the longitudinal study should be applied.

2- Frequentist statistics uses P-value as a measure of probability to show that the effect exists and the observed effect is statistically meaningful; judging about the existence of effect is not subjective and cannot be shown without P-value (or confidence interval), so reporting p-value is not an optional choice.

Of course, there are some drawback to frequentist inference including the p-value approach to hypothesis testing, but it’s the only method for frequentists to test their hypothesis; The ASA statement is informing researchers to be aware of problems related to p-value and warn them to be cautious about the interpretation of the results; but without having the alternative to p-value, it cannot be omitted from the results; if a researcher does not accept the frequentist approach and their use of p-value for hypothesis testing, he/she can apply another branch of statistics, called Bayesian inference, that addresses some limitation of frequents approach including the use of p-value; otherwise, if the frequentist method is used to analyze the data, their rules must be followed exactly.

The Authors referred to a checklist they used to organize their results in order to justify their reported outcome, however, the checklist only mentioned the key points that must be reported in observational studies and does not explain the details of the application of the statistical method and interpretation of the results; for applying the statistical method correctly and vigorously, the authors need to refer to statistical textbooks.

Reviewer #2: Thank you for the good work in addressing my comments which were addressed to my satisfaction. I believe this manuscript will make an important contribution to the ongoing and rapidly developed COVID19 literature.

Reviewer #3: This manuscript adds to the literature health and social behavior changes associated with COVID-19 and provides a unique angle by examining differences in adherence by language regions. The authors addressed reviewer comments appropriately in the manuscript. All data are fully available online without restriction.

7. PLOS authors have the option to publish the peer review history of their article (what does this mean?). If published, this will include your full peer review and any attached files.

Reviewer #1: No

Reviewer #2: No

Reviewer #3: No

---

## [Author Response · Author response to Decision Letter 1]

22 Jul 2021

*** For a better readable form of our response, see the "response to reviewers" file. ***

Re: “Health and social behaviour through pandemic phases in Switzerland: Longitudinal analysis of the COVID-19 Social Monitor panel study”

Winterthur, Juli 22 2021

Dear Prof. Dr. Guidi

We thank you for the opportunity to submit a revised version of our manuscript to PLOS One. Please find below our point-by-point answers to the comments of reviewer #1. Reviewers #2 and #3 had no further comments which need to be addressed in the revised manuscript.

Based on the comment of reviewer #1 we changed the statistical analysis approach towards a hierarchical regression model which accounts for repeated measurements within individuals. Our previous analysis strategy used a cluster-robust approach for addressing repeated measurements which was criticized by reviewer #1. Compared to our previous analyses we found evidence for an interaction effect for the study outcomes poor quality of life (p<0.001), depressive mood (p=0.005), no physical activity (p=0.04), besides the already reported adherence to mitigation measures (p<0.001), from adjusted hierarchical logistic regression models. These findings do not change our previous discussion or interpretation and can likely be explained by the correlation structure between repeated measurements.

Further reviewer #1 required to add p-values for analyses. We added p-values from adjusted regression models (S7 Table).

We updated our discussion about socioeconomic factors with a recently published study which investigated the inverse care law in Switzerland (doi.org/10.1016/S2468-2667(21)00160-2).

We hope that the Editor and reviewer #1 are satisfied with our newly revised manuscript.

Yours sincerely

Marc Höglinger, PhD (on behalf of all co-authors)

PONE-D-21-02911R1

Health and social behaviour through pandemic phases in Switzerland: Regional time-trends of the COVID-19 Social Monitor panel study

PLOS ONE

[ No additional revision points has been raised by the Editor. ]

6. Review Comments to the Author

Reviewer #1: The paper has two main problems that the authors did not address:

First, the statistical method applied inappropriately; second, the statistical interpretation of the results is incorrect.

We appreciate the thoughtful feedback and time efforts of reviewer #1.

1- For investigating the pattern of change during the time, the statistical model must have the ability to investigate the change of outcome during the time, but the logistic regression that was applied in this study is only able to estimate the outcome (odds ratio in this paper) for the specific time point and cannot compare the odds ratios in different time points to show whether the pattern of change exists or not; for showing the pattern of change during the time the longitudinal study should be applied.

Our previous model approach is mentioned in classical statistical textbooks for longitudinal analyses like, for instance, Fixed Effects Regression Methods for Longitudinal Data Using SAS from Paul Allison (ISBN 978-1-59047-568-3), Chapter 3.3. Estimation of Logistic Models for Two or More Observations Per Person. In our previous analysis strategy, we accounted for repeated measurements on individuals with a cluster-robust approach.

Based on the comments from reviewer #1, we changed our modeling strategy to a hierarchical logistic regression model which accounts for repeated measurements on the same individual. That is, we account for nested repeated measurements within individuals while investigating a common fixed effect over periods (Reference: Models for Discrete Longitudinal Data from Molenberghs and Verbeke, doi: 10.1007/0-387-28980-1).

We note that an odds ratio inherently has a reference group, i.e., one compares the odds of the presence of the study outcome of a specific variable category (say, time period 3) with the odds of the reference category (say, time period 1). In our case the reference period is the first period as defined in the manuscript. Thus, our odds ratio estimates are not only for one specific time point – as reviewer #1 says - but compares subsequent time periods with the reference period.

2- Frequentist statistics uses P-value as a measure of probability to show that the effect exists and the observed effect is statistically meaningful; judging about the existence of effect is not subjective and cannot be shown without P-value (or confidence interval), so reporting p-value is not an optional choice.

Of course, there are some drawback to frequentist inference including the p-value approach to hypothesis testing, but it’s the only method for frequentists to test their hypothesis; The ASA statement is informing researchers to be aware of problems related to p-value and warn them to be cautious about the interpretation of the results; but without having the alternative to p-value, it cannot be omitted from the results; if a researcher does not accept the frequentist approach and their use of p-value for hypothesis testing, he/she can apply another branch of statistics, called Bayesian inference, that addresses some limitation of frequents approach including the use of p-value; otherwise, if the frequentist method is used to analyze the data, their rules must be followed exactly.

We added to the statistical section that we defined a 5% alpha level as statistically significant and reported p-values in S7 Table. All other p-values for interactions and time effects have already been reported.

The Authors referred to a checklist they used to organize their results in order to justify their reported outcome, however, the checklist only mentioned the key points that must be reported in observational studies and does not explain the details of the application of the statistical method and interpretation of the results; for applying the statistical method correctly and vigorously, the authors need to refer to statistical textbooks.

As mentioned in the previous rebuttal letter, we based our reasoning to focus on effect size estimation (rather than p-value based assessments of significance) on different statistical textbooks (Essential Medical Statistics by Kirkwood and Sterne, ISBN 978-0-865-42871-3), “Likelihood and Bayesian Inference” from Held and Sabanés Bové (ISBN 978-3-662-60792-3), and “Modern Epidemiology” form Lash, VanderWeele, Haneuse, Rothman (ISBN 978-3-662-60792-3).

Reviewer #2: Thank you for the good work in addressing my comments which were addressed to my satisfaction. I believe this manuscript will make an important contribution to the ongoing and rapidly developed COVID19 literature.

We thank reviewer #2 for the helpful comments and feedback, and highly appreciate the time efforts for the review. We note that we changed the statistical analysis approach towards a hierarchical regression model which accounts for repeated measurements within individuals. Our previous analysis strategy used a cluster-robust approach for addressing repeated measurements. Further, we updated our discussion about socioeconomic factors with a recently published study which investigated the inverse care law in Switzerland (doi.org/10.1016/S2468-2667(21)00160-2).

Reviewer #3: This manuscript adds to the literature health and social behavior changes associated with COVID-19 and provides a unique angle by examining differences in adherence by language regions. The authors addressed reviewer comments appropriately in the manuscript. All data are fully available online without restriction.

Thank you. We believe that your feedback substantially improved our manuscript, and we highly appreciate the time efforts for the review. We note that we changed the statistical analysis approach towards a hierarchical regression model which accounts for repeated measurements within individuals. Our previous analysis strategy used a cluster-robust approach for addressing repeated measurements. Further, we updated our discussion about socioeconomic factors with a recently published study which investigated the inverse care law in Switzerland (doi.org/10.1016/S2468-2667(21)00160-2).

We thank the Editor and the reviewers for their helpful feedback which substantially improved our manuscript.

Marc Höglinger (and Co-authors)

---

## [Decision Letter · Decision Letter 2]

4 Aug 2021

Health and social behaviour through pandemic phases in Switzerland: Regional time-trends of the COVID-19 Social Monitor panel study

PONE-D-21-02911R2

Dear Dr. Höglinger,

We’re pleased to inform you that your manuscript has been judged scientifically suitable for publication and will be formally accepted for publication once it meets all outstanding technical requirements.

Kind regards,

Barbara Guidi

Academic Editor

PLOS ONE

Additional Editor Comments (optional):

Reviewers' comments:

Reviewer's Responses to Questions

**Comments to the Author**

1. If the authors have adequately addressed your comments raised in a previous round of review and you feel that this manuscript is now acceptable for publication, you may indicate that here to bypass the “Comments to the Author” section, enter your conflict of interest statement in the “Confidential to Editor” section, and submit your "Accept" recommendation.

Reviewer #2: All comments have been addressed

2. Is the manuscript technically sound, and do the data support the conclusions?

Reviewer #2: Yes

3. Has the statistical analysis been performed appropriately and rigorously? 

Reviewer #2: Yes

4. Have the authors made all data underlying the findings in their manuscript fully available?

Reviewer #2: No

5. Is the manuscript presented in an intelligible fashion and written in standard English?

Reviewer #2: Yes

6. Review Comments to the Author

Reviewer #2: My comments had been addressed to my satisfaction. As before, I believe this paper will make an important contribution to this rapidly developing knowledge. Thank you

7. PLOS authors have the option to publish the peer review history of their article (what does this mean?). If published, this will include your full peer review and any attached files.

Reviewer #2: No

---

## [Editor Report · Acceptance letter]

11 Aug 2021

PONE-D-21-02911R2 

Health and social behaviour through pandemic phases in Switzerland: Regional time-trends of the COVID-19 Social Monitor panel study 

Dear Dr. Höglinger:

I'm pleased to inform you that your manuscript has been deemed suitable for publication in PLOS ONE. Congratulations! Your manuscript is now with our production department. 

Kind regards, 

on behalf of

Dr. Barbara Guidi 

Academic Editor

PLOS ONE